# Focused ultrasound excites cortical neurons via mechanosensitive calcium accumulation and ion channel amplification

Sangjin Yoo[1], David R. Mittelstein[2], Robert C. Hurt [3], Jerome Lacroix [1,4] & Mikhail G. Shapiro [1✉]

Ultrasonic neuromodulation has the unique potential to provide non-invasive control of neural activity in deep brain regions with high spatial precision and without chemical or genetic modification. However, the biomolecular and cellular mechanisms by which focused ultrasound excites mammalian neurons have remained unclear, posing significant challenges for the use of this technology in research and potential clinical applications. Here, we show that focused ultrasound excites primary murine cortical neurons in culture through a primarily mechanical mechanism mediated by specific calcium-selective mechanosensitive ion channels. The activation of these channels results in a gradual build-up of calcium, which is amplified by calcium- and voltage-gated channels, generating a burst firing response. Cavitation, temperature changes, large-scale deformation, and synaptic transmission are not required for this excitation to occur. Pharmacological and genetic inhibition of specific ion channels leads to reduced responses to ultrasound, while over-expressing these channels results in stronger ultrasonic stimulation. These findings provide a mechanistic explanation for the effect of ultrasound on neurons to facilitate the further development of ultrasonic neuromodulation and sonogenetics as tools for neuroscience research.

[1] Division of Chemistry and Chemical Engineering, California Institute of Technology, Pasadena, CA 91125, USA. [2] Division of Engineering and Applied Science, California Institute of Technology, Pasadena, CA 91125, USA. [3] Division of Biology and Biological Engineering, California Institute of Technology, Pasadena, CA 91125, USA. [4] Graduate College of Biomedical Sciences, Western University of Health Sciences, Pomona, CA 91766, USA. ✉email: mikhail@caltech.edu

Non-invasive neuromodulation technologies play a critical role in basic neuroscience research and the development of therapies for neurological and psychiatric disease. However, established non-invasive techniques such as transcranial magnetic stimulation (TMS) and transcranial direct current stimulation (tDCS) suffer from limited spatial targeting and penetration depth[1]. In contrast, focused ultrasound (FUS) has the potential to modulate neural activity in deep-brain regions with millimeter spatial precision based on the penetrance of sound waves in bone and soft tissue. Recently, transcranial FUS in the frequency range of 0.25–1 MHz and intensity of 1–100 W/cm$^2$ ($I_{SPPA}$) has been shown to elicit neural and behavioral responses in small[2–7] and large[8–14] model animals and humans[15–19] without genetic or chemical alterations or deleterious side effects, even with chronic stimulation[20]. These studies have driven widespread interest in the development of FUS as a research tool in neuroscience and a strategy for disease treatment[21,22].

Despite the intense interest in this technology, the underlying cellular and molecular mechanisms of ultrasonic neuromodulation are largely unknown. The study of these mechanisms is made challenging as ultrasound produces multiple physical effects, including mechanical force, heating, and cavitation[23–29]. The role of these physical processes in neuromodulation and their transduction to molecular signals in neurons have not been elucidated. Moreover, recent findings of off-target auditory effects of FUS in small animals make it challenging to study potential mechanisms in the in vivo context[30,31]. While several theoretical and experimental proposals have been advanced[23–25,28,29,32–35], no consensus exists about how ultrasound modulates neuronal activity at the molecular and cellular level[22].

Here, we describe a comprehensive study of the molecular and cellular mechanisms of ultrasonic neuromodulation in primary cortical neurons. Using stimulation and readout methods consistent with the acoustic conditions expected in vivo, we first narrow down the biophysical basis by which ultrasound excites neurons, observing no involvement of temperature elevation, cavitation or large-scale deformation. Then, we uncover a signaling pathway whereby the mechanical effects of ultrasound cause calcium influx through specific endogenous mechanosensitive ion channels. We find that this triggers signal amplification by calcium-gated sodium channels, and ultimately results in robust spiking activity. This pathway functions internally within neurons and does not require synaptic transmission. The overexpression of specific mechano-sensitive and amplifier channels identified in our biophysical experiments significantly enhances ultrasound response magnitude and kinetics. These results provide comprehensive mechanistic insights into the excitatory action of ultrasound on mammalian neurons, with important implications for the development of ultrasonic neuromodulation and sonogenetics.

## Results

### Focused ultrasound robustly activates cortical neurons. 
To study neuronal responses to focused ultrasound under acoustic conditions matching soft tissue, we cultured primary murine cortical neurons on an acoustically transparent mylar film while optically recording their calcium and voltage responses to ultrasound using genetically encoded fluorescent indicators (Fig. 1a). The neurons were placed at the top of a water tank, with a focused ultrasound transducer submerged in degassed water below them and angled to reduce standing wave formation. The 5 mm focal diameter of the transducer (Fig. 1b) delivered ultrasound uniformly to neurons throughout our field of view. We used a frequency of 300 kHz, within the range utilized in recent studies in a variety of organisms[9,11,14–19,36], and continuous-wave stimulation, which was found to be as effective as pulsed ultrasound[4]. The inter-pulse interval was fixed at 20 s to allow a return to baseline.

To establish the pulse parameters under which neurons in culture respond to ultrasound, we stimulated the cells across a range of pulse intensities (0–15 W/cm$^2$) and pulse durations (0–500 ms, CW, continuous wave) while imaging calcium responses with virally transfected GCaMP6f (Fig. 1c and Supplementary Video 1). The neurons showed robust responses, with amplitudes increasing monotonically with intensity and pulse duration (Fig. 1d, e). The magnitude of these responses was larger than those produced by the neurons' spontaneous spiking activity (Supplementary Fig. 1b). The calcium response was not immediate, but had a delay of approximately 200 ms after the onset of stimulation (Fig. 1f). This onset delay is not explained by the rise time of GCaMP6f fluorescence (time to peak ~45 ms)[37], and therefore reflects the kinetics of the neurons' response to FUS. Both this onset delay and the maximum response time (~1.7 s), were reduced significantly by increasing ultrasound intensity (Fig. 1g, h). We also stimulated cells with pulsed wave (1 KHz and 1.5 KHz PRF, pulse repetition frequency) and higher frequency (670 KHz) ultrasound and found no substantial differences in response amplitude or onset delay (Supplementary Fig. 1).

Based on these results, we set our subsequent stimulation parameters to 15 W/cm$^2$ and 500 ms (CW), which are similar to those used in large animal and human studies[9,11,14–19,36]. To ensure that these ultrasound parameters were not damaging to cells, we looked for and found neither sustained calcium accumulation nor irreversible membrane perforation after repetitive stimuli (Supplementary Fig. 2).

To determine whether the observed responses to FUS were specific to the adherent 2D culture format, we also applied ultrasound to neurons in 3D collagen culture. We found that neurons in this format also showed reliable calcium signals after the onset delay in response to stimulation (Supplementary Fig. 3). In addition, to determine the extent to which standing waves, which are nearly impossible to completely eliminate in a FUS setup in vitro and in vivo[38,39], play a role in the observed excitation, we applied FUS to neurons with a chirped waveform, which ameliorates the pressure gradients induced by standing waves. We found the calcium signal unaffected in terms of response amplitude and response delay (Supplementary Fig. 4).

### Ultrasound excites neurons via mechanical force. 
Focused ultrasound is capable of producing multiple physical phenomena in tissue, including elevating temperature, inducing bubble formation and cavitation, and applying mechanical force (Fig. 2a), each of which could potentially lead to neuronal excitation[24,28,29,40,41]. To determine which of these phenomena are involved in stimulating cortical neurons, we first measured changes in temperature during FUS application. A fiber optic thermometer positioned adjacent to the neurons recorded temperature changes of 0.005 ± 0.003 °C in response to our optimized ultrasound parameters (Fig. 2b), and changes below 0.02 °C at all parameters tested using a 300 kHz transducer (Supplementary Fig. 5a). The absence of a major temperature increase was corroborated by co-expressing the fluorescent protein mCherry[42] as a temperature indicator (Supplementary Fig. 5b), and showing that the mCherry fluorescence stayed constant while neurons responded to ultrasound (Fig. 2c). Although these assays measure bulk temperature in media proximal to the cells or inside the cytoplasm rather than locally within the cell membrane, Fourier's law of heat diffusion predicts a thermal equilibration length scale on the order of 100 μm in aqueous media[43] during the 100 ms timescale of our observed responses. Our results thus suggest that

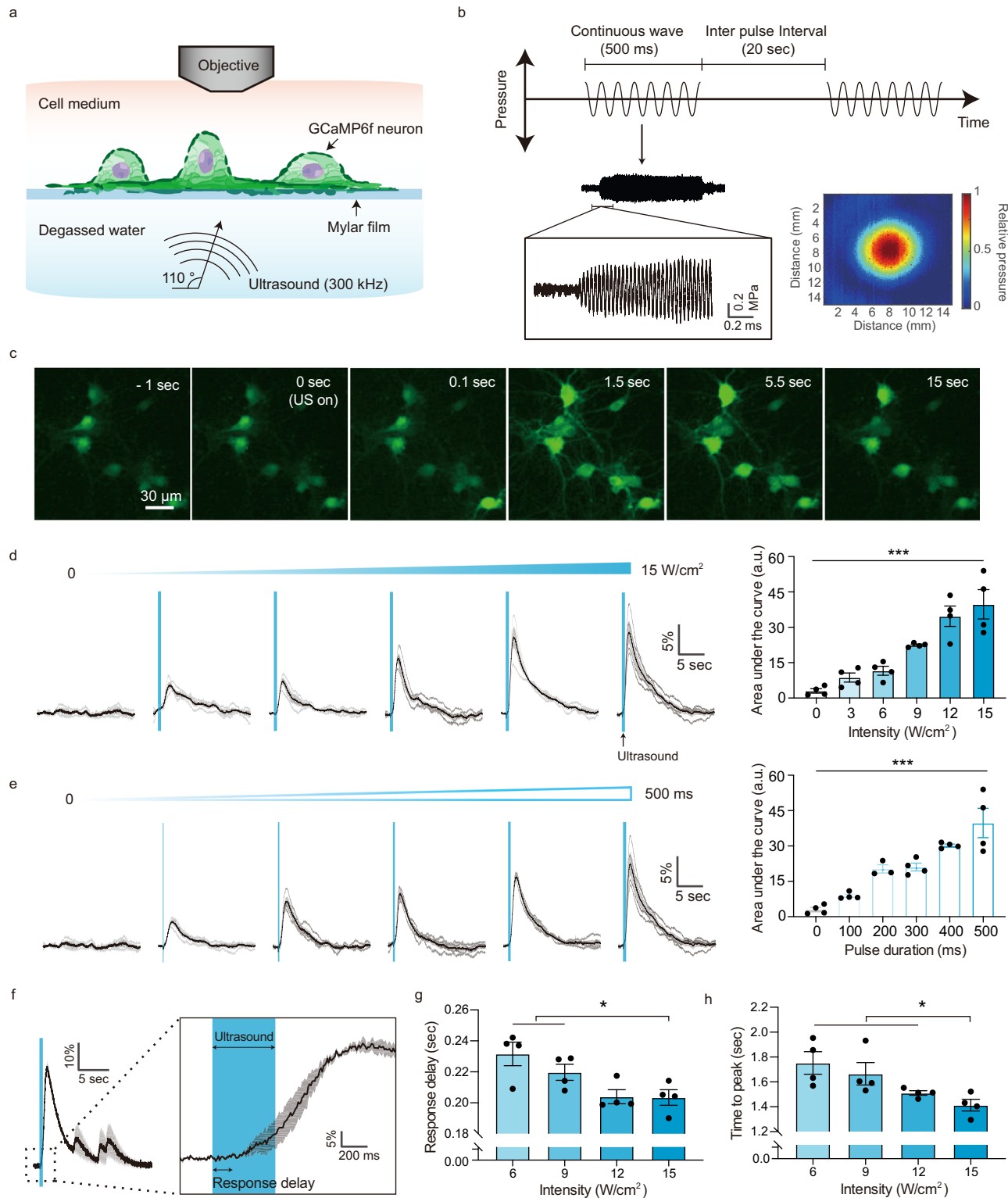

temperature is unlikely to play a major role in ultrasonic neuromodulation in this parameter range, as predicted by numerical estimates[44,45].

Among the potential non-thermal effects of ultrasound, bubble formation and cavitation have been hypothesized as a mechanism for ultrasonic neuromodulation due to the observation of enhanced responses at lower frequencies[23]. To assess the relevance of this phenomenon in cultured neurons, we compared their responses to ultrasound in atmospherically gassed and degassed cellular media, with the latter condition disfavoring cavitation. No significant differences were observed (Fig. 2d). In addition, we looked for bubbles directly using an ultra-high-speed camera (5 MHz frame rate), and saw no bubbles formed in the vicinity of neurons during FUS application (Fig. 2e and Supplementary Fig. 5c). Images were recorded starting 100 ms after the onset of FUS, providing sufficient time for bubble growth[24] and approaching the latency of our observed neuronal excitation. These results are consistent with our mechanical index

**Fig. 1 Cultured cortical neurons are excited by focused ultrasound stimulation. a** Illustration of the focused ultrasound stimulation setup. Angled ultrasound waves (300 kHz) are delivered to GCaMP6f-expressing neurons cultured on an acoustically transparent mylar film, while the neural calcium response is recorded by epifluorescence imaging. **b** Schematic of the acoustic waveform applied to neurons and representative focal pressure waveform measured by a hydrophone. Temporal offset from the rectangular waveform (signal from function generator) reflects the focal length of the transducer. The colormap shows the spatial profile of the acoustic pressure at the ultrasound focus, with a full-width at half-maximal diameter 5.2 mm. **c** Representative time lapse images of GCaMP6f fluorescence before, during and after an ultrasound stimulation ($15\ \text{W/cm}^2$, 500 ms pulse duration at 0 s). **d** Calcium responses and quantification of neural response as function of ultrasound intensity ($n = 4$ independent experiments each, one-way ANOVA $p < 0.0001$) and **e** pulse duration ($n = 4$ independent experiments each, one-way ANOVA $p < 0.0001$). **f** A representative single cell response to ultrasound. **g** Quantification of response onset time ($n = 4$ independent experiments each, one-way ANOVA $p = 0.0117$, Tukey's post comparison) and **h** time to peak ($n = 4$ independent experiments each, one-way ANOVA $p = 0.0190$, Tukey's post comparison). Individual traces are gray solid, their mean trace is black solid, and SEM is shaded. Bar graph values represent mean ± SEM.

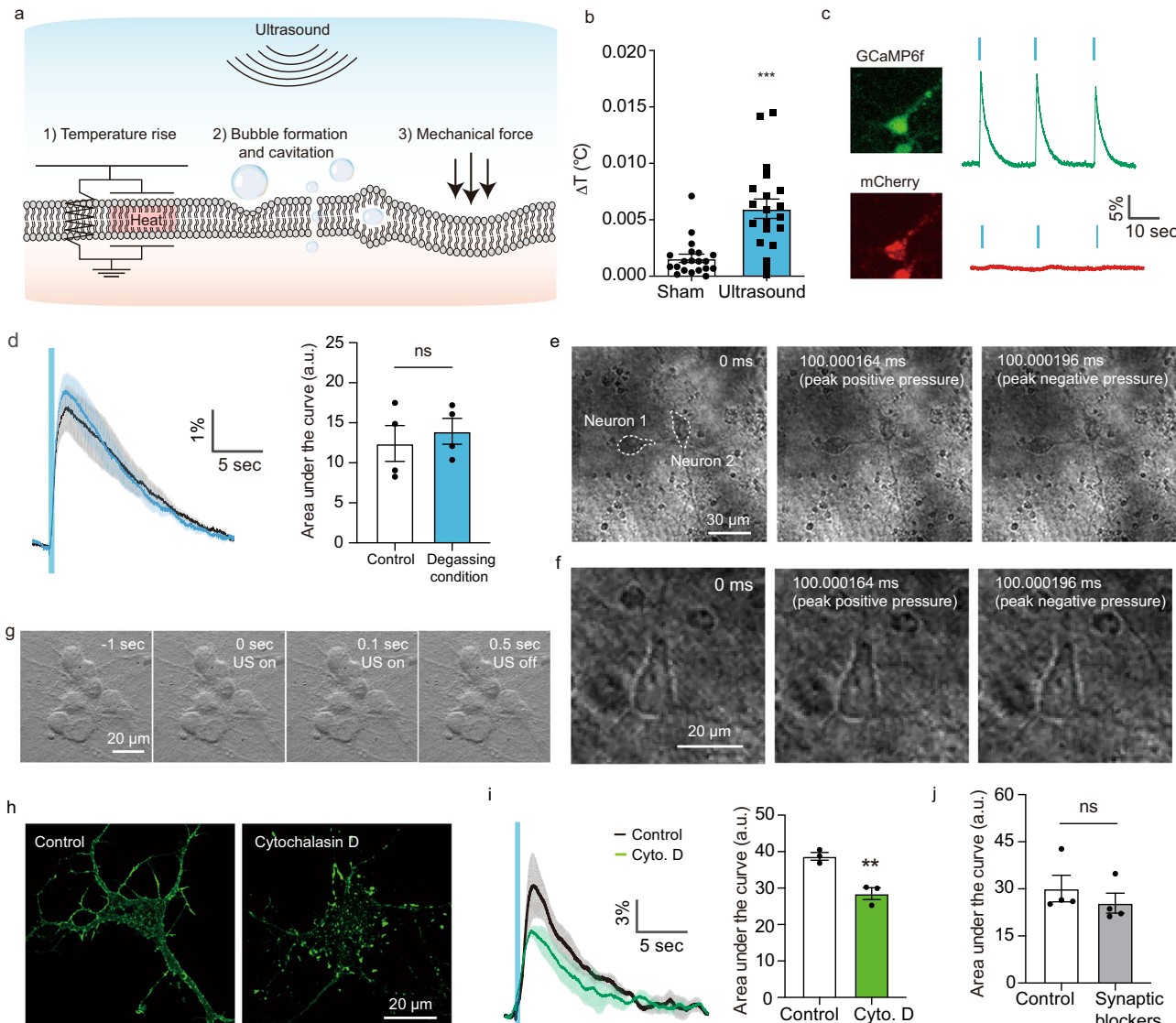

**Fig. 2 Ultrasound excites neurons through direct mechanical effects. a** Illustration of the potential biophysical effects of ultrasound. **b** Temperature increase measured using an optic hydrophone thermometer positioned near the neurons during ultrasound stimulation ($n = 20$, $15\ \text{W/cm}^2$, 500 ms pulse duration with 20 s inter-pulse interval, Unpaired $T$-test, two-tailed, $p < 0.0001$). **c** Fluorescence images of a neuron co-expressing GCaMP6f (green) and mCherry (red) and changes in their respective fluorescence in response to ultrasound stimulation. **d** Calcium responses to ultrasound in freshly degassed media ($n = 4$ independent experiments each, unpaired $t$-test, two-tailed, $p = 0.6033$). **e** Ultra-high-speed imaging (5 Mfps) of neurons and surrounding media during ultrasound stimulation. Image recording was started 100 ms after the onset of ultrasound. **f** Ultra-high-speed imaging of a single neuron during ultrasound stimulation at higher magnification. **g** Bright field imaging of neurons over the full time course of the ultrasound stimulation. **h** Images of individual neurons with the F-actin label Alexa-Fluor 488 phalloidin before and after treatment with cytochalasin D. **i** Calcium responses before and after cytochalasin D treatment, and quantification of area under the curve ($n = 3$ independent experiments, unpaired $t$-test, two-tailed, $p = 0.0061$). **j** Quantification of area under the curve after applying the synaptic blockers AP5 and CNQX (1 µM each, $n = 4$ independent experiments, paired $T$-test, two-tailed, $p = 0.4128$). Mean trace is solid and SEM is shaded. Bar graph values represent mean ± SEM.

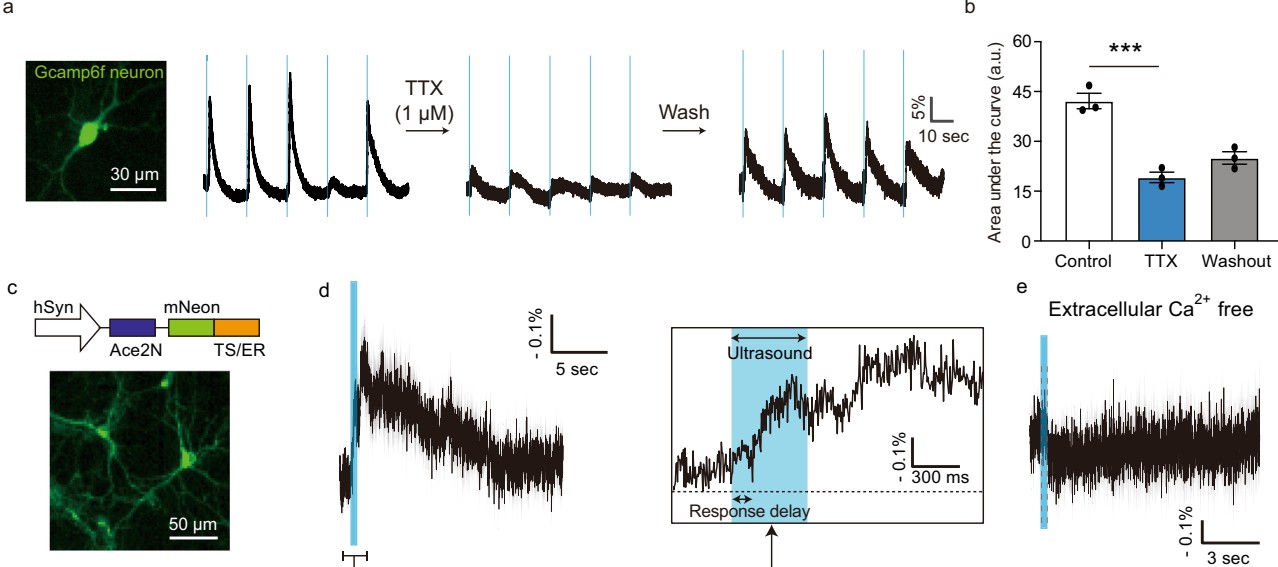

**Fig. 3 Ultrasound response is mediated by the entry of extracellular calcium. a** Calcium responses from a single neuron during ultrasound stimulation before, during and after treatment with the sodium channels blocker tetrodotoxin (TTX). **b** Quantification of area under the curve before, during and after TTX treatment ($n = 3$ independent experiments, one-way ANOVA $p = 0.0004$, Tukey's post comparison (control vs. TTX)). **c** Diagram of the Ace2N voltage indicator genetic construct and representative fluorescence image of neurons transfected with this construct. **d** Voltage responses to ultrasound. ($n = 4$ independent experiments). **e** Voltage responses of neurons in calcium-free media ($n = 2$ independent experiments). Mean trace is solid and SEM is shaded. Bar graph values represent mean ± SEM.

(MI = 0.9) being below the typical cavitation threshold in soft tissue (MI > 1.9)[46]. Although the optical resolution of our setup (~532 nm) was insufficient to directly visualize the formation of the nanoscale bubbles hypothesized by some theoretical studies to form inside the membrane bilayer[24,28,29], the gross appearance of the membrane remained unchanged over the ultrasound cycle (Fig. 2f). This suggests that there were no major changes in refractive index, as might be expected if a large fraction of the membrane surface undergoes cavitation, as required by the intramembrane cavitation theory[24,28,29]. The voltage implications of the theory are further examined in the next section of the manuscript.

Having ruled out temperature changes and cavitation, we next focused on direct mechanical forces. Given the similar acoustic impedance of neurons and surrounding media and the absence of large pressure gradients within our ultrasound focus, the acoustic radiation force on the neurons due to FUS is expected to be weak[47,48], with any resulting deformations expected to be below our optical detection limit. Indeed, under high-speed imaging we observed no significant cell deformation either during each wave cycle (Fig. 2f and Supplementary Fig. 5d) or over the longer course of the ultrasound pulse (Fig. 2g). While we did not perform ultra-high-speed imaging in the FUS application period before 100 ms, any deformation due to acoustic radiation force is expected to increase and persist on this timescale. As an indirect test of the involvement of mechanical deformation in the neuromodulation response, we altered the mechanical properties of the neurons by depolymerizing their actin cytoskeleton, which plays a critical role in establishing the elastic modulus of the cytoplasm and cellular cortex. When we depolymerized actin using cytochalasin D[49], at concentrations that did not affect spontaneous excitability or viability of the neurons (Fig. 2h and Supplementary Fig. 5e)[49,50], we observed a significant reduction in the amplitude of the evoked calcium response (Fig. 2i). This suggests that mechanical stress is involved in ultrasonic neuromodulation, albeit in a manner not resulting in, or requiring, micron-scale deformation of the cell.

As a final question before delving into molecular mechanisms, we asked whether the neuronal response to ultrasound was cell-autonomous or required synaptic connections with excitatory neurons[25] or astrocytes[32]. After treating the neurons with the postsynaptic blockers AP5 and CNQX, we found that the neuronal response to ultrasound was not greatly affected (Fig. 2j), suggesting that synaptic transmission is not required for excitation. Synaptic transmission was shown to play a role in previous experiments[25], and a small effect could not be ruled out by our results.

**Ultrasound stimulation triggers calcium entry across the plasma membrane.** To determine the molecular basis of the neuronal response to ultrasound, we first examined which ions enter the cell during FUS stimulation. Since neurons contain voltage-gated $Ca^{2+}$ channels, it is not possible to determine from GCaMP responses alone whether $Ca^{2+}$ enters the cell directly as a result of ultrasound or due to action potential firing. However, consistent with previous results in slices[25], blocking voltage-gated sodium channels with TTX only partially reduced the magnitude of the ultrasound response (Fig. 3a, b and Supplementary Fig. 1a). This suggests that calcium enters the cell directly as a result of ultrasound application, in addition to its entry following depolarization. To confirm the role of $Ca^{2+}$ as a primary initiator of the response to FUS, we imaged transmembrane voltage using the genetically encoded voltage indicator Ace2N (Fig. 3c). In the normal, calcium-containing media, FUS application elicited depolarizations (Supplementary Fig. 6a), with a response onset similar to that observed with GCaMP6f (Fig. 3d). In contrast, in calcium-free media this voltage response to ultrasound was completely eliminated (Fig. 3e), while the cells retained their ability to respond to other stimuli (Supplementary Fig. 6b). These results demonstrate that extracellular $Ca^{2+}$ is the essential ionic initiator of ultrasonic neuromodulation. Intracellular calcium release from the endoplasmic reticulum does not play a major additional role (Supplementary Fig. 7).

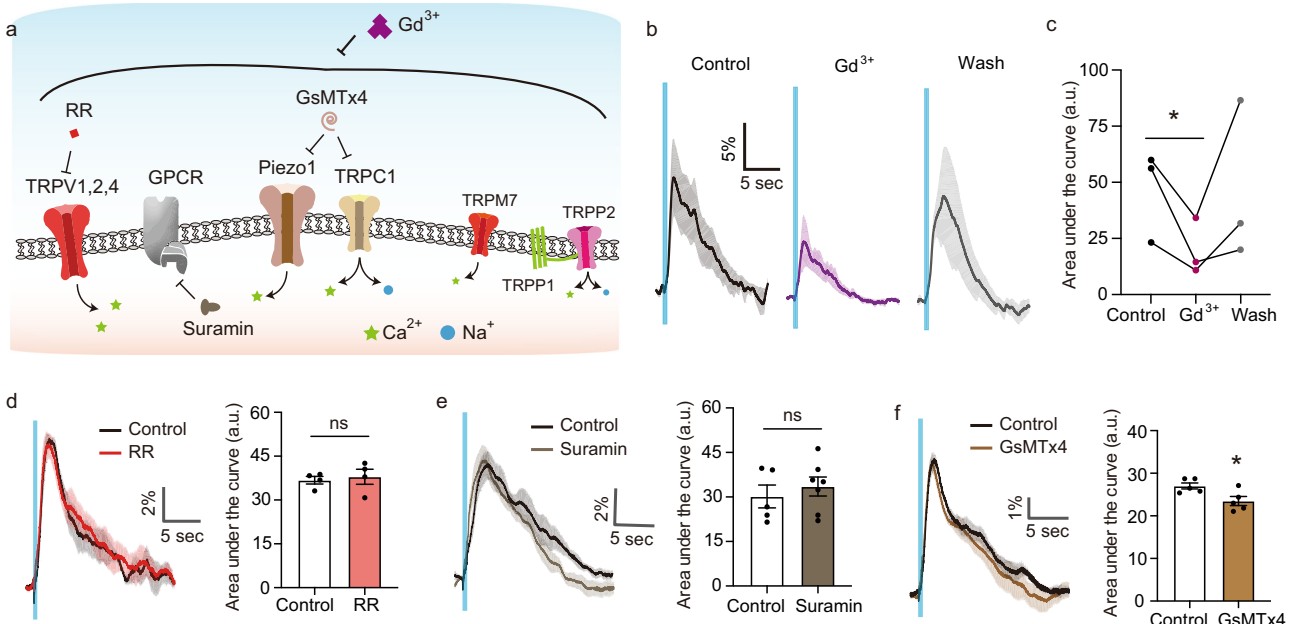

**Fig. 4 Pharmacological inhibition of mechanosensitive receptors. a** Schematic of neuronal mechanosensitive receptors and strategies to block them. Gadolinium ($Gd^{3+}$, 20 μM) was used to block the mechanosensitive channels nonspecifically. Pores of TRPV1, 2 and 4 channels were blocked by ruthenium red (RR, 1 μM). Activation of GPCRs was inhibited by suramin (60 μM). Gating of Piezo1 and TRPC1 channels was inhibited by GsMTx4 (10 μM). **b** Calcium responses before, during and after treatment with $Gd^{3+}$. **c** Average calcium response under each condition ($n = 3$ independent experiments, Paired $T$-test, two-tailed, $p = 0.0884$). **d** Calcium responses before and after treatment with RR ($n = 3$ independent experiments, unpaired $t$-test, $p = 0.6930$). **e** Calcium responses before and after treatment with suramin ($n = 5$ independent experiments for control, and seven independent experiments for suramin, unpaired $t$-test, two-tailed, $p = 0.5159$). **f** Calcium responses before and after treatment with GsMTx4 ($n = 5$ independent experiments each, unpaired $t$-test, two-tailed, $p = 0.0245$). Mean trace is solid and SEM is shaded. Bar graph values represent mean ± SEM.

Voltage imaging also provided an additional method to test the intramembrane cavitation theory, which hypothesizes that bubble formation leads to rapidly oscillating hyperpolarizing currents, resulting in action potential generation through a charge accumulation mechanism[24,28,29]. Although the kinetics of our voltage sensor are not fast enough to capture membrane potential oscillation at the ultrasound frequency, we would expect it to pick up time-averaged hyperpolarization during ultrasound application. However, no such hyperpolarization was observed (Fig. 3d), and this result was corroborated in spiking HEK cells as a generic excitable membrane model (Supplementary Fig. 6c, d).

**Ultrasound stimulation activates specific mechanosensitive ion channels**. Having established that ultrasound excites neurons via mechanical force resulting in the entry of extracellular calcium, we hypothesized that this response involves the activation of endogenous mechanosensitive ion channels[16,23,33,51]. Cortical neurons have been shown to express multiple channels with reported mechanosensitivity, including TRPV1, TRPV2, TRPV4, Piezo1, TRPC1, TRPM7 and the TRPP1/2 complex[51,52]. Mechanosensitive ionic currents can also be mediated indirectly by G-protein coupled receptors (GPCRs)[53]. To determine which channels are involved in ultrasonic neuromodulation, we first blocked subsets of candidate mechanosensitive receptors using pharmacological blockers, then used CRISPR/Cas9 knockdown to further delineate the roles of specific proteins (Fig. 4a).

We started by treating the neurons with gadolinium(III), which modifies the deformability of the lipid bilayer[54], resulting in changes in membrane mechanics leading to inhibition of mechanosensitive ion channels. The dose of $Gd^{3+}$ was carefully chosen to avoid blocking non-mechanosensitive channels or otherwise altering cell excitability[55] (Supplementary Fig. 8a). In the presence of $Gd^{3+}$, the amplitude of the evoked responses was significantly reduced, decreasing by 60% (Fig. 4b, c). This confirmed that mechanosensitive channels are involved in ultrasound transduction. The incomplete elimination of the FUS response may be due to our use of a relatively low dose of $Gd^{3+}$ to avoid non-specific effects, and possibly the involvement of mechanosensitive channels gated by mechanisms not requiring bilayer deformation[56]. Response kinetics were not significantly affected (Supplementary Fig. 8f).

Next, we used selective chemical blockers to inhibit distinct mechanosensitive channels, carefully titrating each drug to avoid non-specific excitability reduction or cytotoxicity or response kinetics (Supplementary Fig. 8b–e). First, we used ruthenium red ($IC_{50}$ ~500 nM, used at 1 μM) to block TRPV1, TRPV2, and TRPV4 channels[57]. The resulting neural responses were not significantly different from controls (Fig. 4d), suggesting that these channels are not involved. Next, we used suramin, which blocks GPCR signaling by inhibiting the release of GDP from the G alpha subunit[58] ($IC_{50}$ ~200 nM, used at 60 μM). Neurons treated with this compound showed no significant change in their response to ultrasound compared to controls (Fig. 4e), suggesting that GPCRs are not involved. We then tested the involvement of the Piezo1 and TRPC1 channels using the peptide inhibitor GsMTx4, which inserts into the stressed membrane and distorts membrane tension near the channels[59,60]. Neurons treated with GsMTx4 ($IC_{50}$ ~5 μM, used at 10 μM), showed a modest but significant reduction in the magnitude of their response to ultrasound (Fig. 4f). This indicates the partial involvement of Piezo1 and/or TPRC1 channels in ultrasonic neuromodulation. Combinations of pharmacological blockers were not tested due to the potential combinatorial effects of the compounds and their co-solvents on neurons' viability and excitability. None of the blockers significantly altered the response delay (Supplementary Fig. 8g–i).

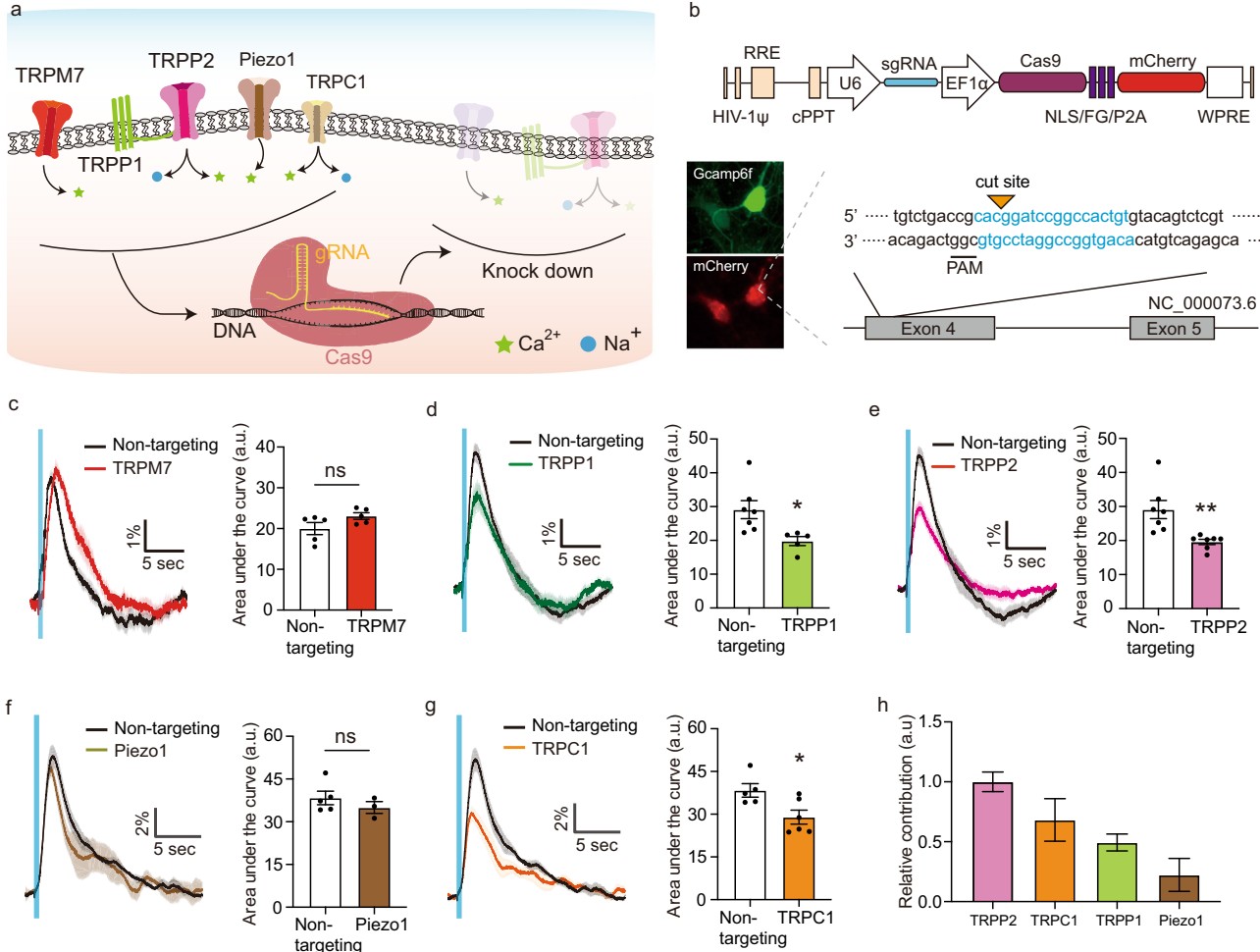

**Fig. 5 CIRSPR/Cas9 knockdown of mechanosensitive ion channels. a** Schematic of the strategy to knockdown individual ion channels using CRISPR/Cas9. **b** Schematic of the gene construct for the CRISPR knockdown. A sgRNA was designed to target each channel and delivered to neurons via lentivirus. **c** Calcium responses from wild-type neurons and neurons treated with CRISPR/Cas9 for TRPM7 knock down (n = 5 independent experiments each, Unpaired t-test, two-tailed, p = 0.1073). **d** Calcium responses from wild type neurons and modified neurons with CRISPR/Cas9 for TRPP1 knock down (n = 7 independent experiments for control, and five independent experiments for TRPP1, Unpaired t-test, two-tailed, p = 0.0208). **e** Calcium responses from wild type neurons and modified neurons with CRISPR/Cas9 for TRPP2 knock down (n = 7 independent experiments each, Unpaired t-test, two-tailed, p = 0.0084). **f** Calcium responses from wild type neurons and modified neurons with CRISPR/Cas9 for Piezo1 knock down (n = 5 independent experiments for control, and three independent experiments for Piezo1, Unpaired t-test, two-tailed, p = 0.3727). **g** Calcium responses from wild type neurons and modified neurons with CRISPR/Cas9 for TRPC1 knock down (n = 5 independent experiments for control, and 6 independent experiments for TRPC1, Unpaired t-test, two-tailed, p = 0.0232). **h** Relative contribution of each channel to the ultrasound-evoked calcium response (normalized ΔΔF/CRISPR efficiency, n = 7 (TRPP2), 6 (TRPC1), 5 (TRPP1), 3 (Piezo1)). Control is non-targeting sgRNA. Mean trace is solid and SEM is shaded in time courses. Bar graph values represent mean ± SEM.

Because selective pharmacological inhibition was not available for all the candidate channels, we also used CRISPR/Cas9 to knock down several channels (Fig. 5a). For each channel, sgRNA sequences were designed using the CRISPRko tool[61] to maximize targeted Cas9 activity and minimize off-target effects. The designed sgRNA was inserted into an all-in-one vector, containing a single sgRNA expression cassette and a Cas9 nuclease expression cassette, and delivered to neurons via lentivirus[62] (Fig. 5b). When sgRNAs were used to target TRMP7, TRPP1, TRPP2, Piezo1, and TRPC1, they produced editing efficiencies of 20.0–39.6%, as quantified by decomposing the target sequence traces (Supplementary Fig. 9a)[63]. The effect of the partial knockdown of each channel was measured by plotting the average calcium response and quantifying the change in the magnitude of the response.

A control (non-targeting sgRNA) produced no significant changes relative to untreated neurons in these response metrics (Supplementary Fig. 9b). The knockdown of TRPM7 also did not

have any significant effect on the response of neurons to ultrasound (Fig. 5c). In contrast, the partial knockdown of TRPP1 and TRPP2 resulted in significant changes in the magnitude of the ultrasound response (Fig. 5d, e). The CRISPR knockdown of Piezo1 did not result in a statistically significant change in the calcium signal, with the results showing a trend toward minor reduction (Fig. 5f). The knockdown TRPC1 resulted in a significant reduction in response (Fig. 5g). Based on response reduction and CRIPSR knockdown efficiency for each channel, we can estimate their relative contributions to the ultrasound response of neurons, showing TRPP2 and TRPC1 to be the most important (Fig. 5h). No significant effects on baseline excitability were observed (Supplementary Fig. 9c, d). Ultrasound response kinetics were not significantly affected (Supplementary Fig. 9e–i). Taken together, these results implicate the TRPP1/2 complex and TRPC1 mechanoreceptors in the neuronal response to ultrasound.

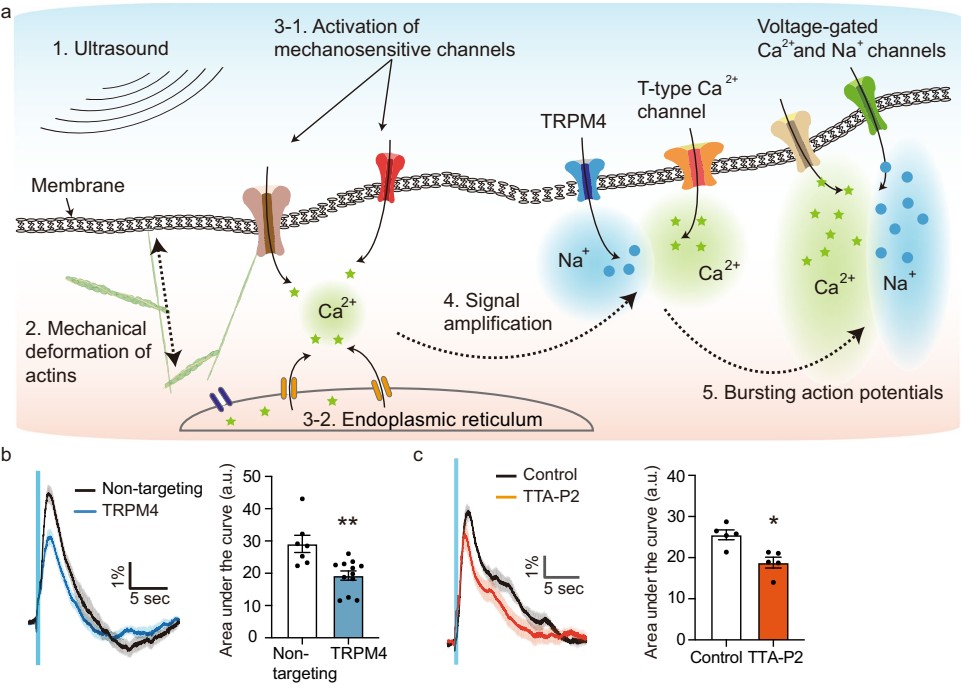

**Fig. 6 Neuronal response to ultrasound is amplified by calcium-gated and voltage-gated ion channels. a** Illustration of the molecular pathway activated by ultrasound. **b** Calcium responses from wild type neurons and modified neurons with CRISPR/Cas9 for TRPM4 knock down ($n = 7$ independent experiments for control, and 12 independent experiments for TRPM4, Unpaired $T$-test, two-tailed, $p = 0.0024$). **c** Calcium responses before and after treatment with the t-type calcium channel blocker TTA-P2 (3 µM, $n = 5$ independent experiments, paired $T$-test, two-tailed, $p = 0.0331$). Mean trace is solid and SEM is shaded. Bar graph values represent mean ± SEM.

**Response to ultrasound is amplified by calcium-gated and low-threshold ion channels**. Since our calcium and voltage imaging experiments indicated that calcium entry was only the initial step in neuronal excitation, we endeavored to further examine the connection between mechanosensitive channel currents and the seconds-long response of the neurons to ultrasound (Fig. 6a). In particular, we focused on the potential role of TRPM4, a non-selective cation channel expressed in cortical neurons, which is activated by intracellular $Ca^{2+}$ at concentrations of 3 µM and facilitates the amplification of small $Ca^{2+}$ signals to larger depolarizing currents[64,65]. We tested the involvement of TRPM4 in the ultrasound response by knocking it down with CRISPR/Cas9, as described above, with an efficiency of 43.4% (Supplementary Fig. 9a). Strikingly, we observed a major reduction in the response magnitude (Fig. 6b), strongly implicating this channel in the ultrasound response pathway. No effects on baseline excitability were observed due to this knockdown (Supplementary Fig. 9c).

Another set of potential downstream amplifiers are the voltage-gated T-type calcium channels, which play an important role in triggering low-threshold spiking and action potential bursting[66]. To test the involvement of this channel class, we treated the cells with the selective pore blocker TTA-P2[67] ($IC_{50} \approx 22$ nM, used at 3 µM, Supplementary Fig. 8e). We observed a significant reduction in the amplitude of calcium responses (Fig. 6c). This result implicates the T-type calcium channels in generating the large and relatively long-lasting responses to ultrasound seen in our preparation and in animal studies[4,8,68]. None of the manipulations significantly altered the response delay (Supplementary Fig. 10).

**Overexpression of mechanosensitive channels and amplifiers enhances the neuronal response to ultrasound**. To test our understanding of the molecular pathways underlying the ultrasound response and facilitate the development of sonogenetic strategies to sensitize genetically defined subsets of neurons to

ultrasound, we overexpressed three of the ion channels identified in our knockdown experiments as having a role in this phenomenon. Based on their smaller size and ability to be packaged in lentiviral transfection vectors, we selected TRPC1, TRPP2, and TRPM4 as representative mechanoreceptors and calcium-dependent amplifier. Each gene was overexpressed in neurons under a hSyn promoter, as confirmed by immunofluorescent labeling (Fig. 7a). No or minor effects on baseline excitability were observed due to this overexpression (Supplementary Fig. 11a–c). Strikingly, ultrasound stimulation of neurons overexpressing TRPC1 and TRPP2 elicited substantially larger calcium responses compared to wild-type cells, and enabled stronger activation at lower ultrasound intensities (Fig. 7b–d).

Likewise, neurons overexpressing the TRPM4 channel showed marked increases in their response amplitude (Fig. 7b, e). These increases were significant under weak to high ultrasound intensity ($\geq 3$ W/cm²). In addition, the overexpression of TRPM4 accelerated the kinetics of the ultrasound response, reducing the onset time of the calcium signal to below 100 ms at 15 W/cm² (Fig. 7f), consistent with previous overexpression studies[64]. No significant changes in the onset kinetics were observed after overexpressing TRPC1 and TRPP2.

In contrast to these three channels, overexpression of TRPV1, which our inhibition experiments showed to be uninvolved in the neuronal response to ultrasound (Fig. 4d), produced no significant change in the neurons' ultrasound-elicited activation (Supplementary Fig. 11d, e). Taken together, these results confirm the roles of TRPC1, TRPP2, and TRPM4 in the neuronal response to ultrasound and suggest that the overexpression of these channels can be used to sensitize neurons to this form of stimulation.

## Discussion

The results of this study provide a detailed biophysical and molecular description of the mechanisms by which ultrasound can

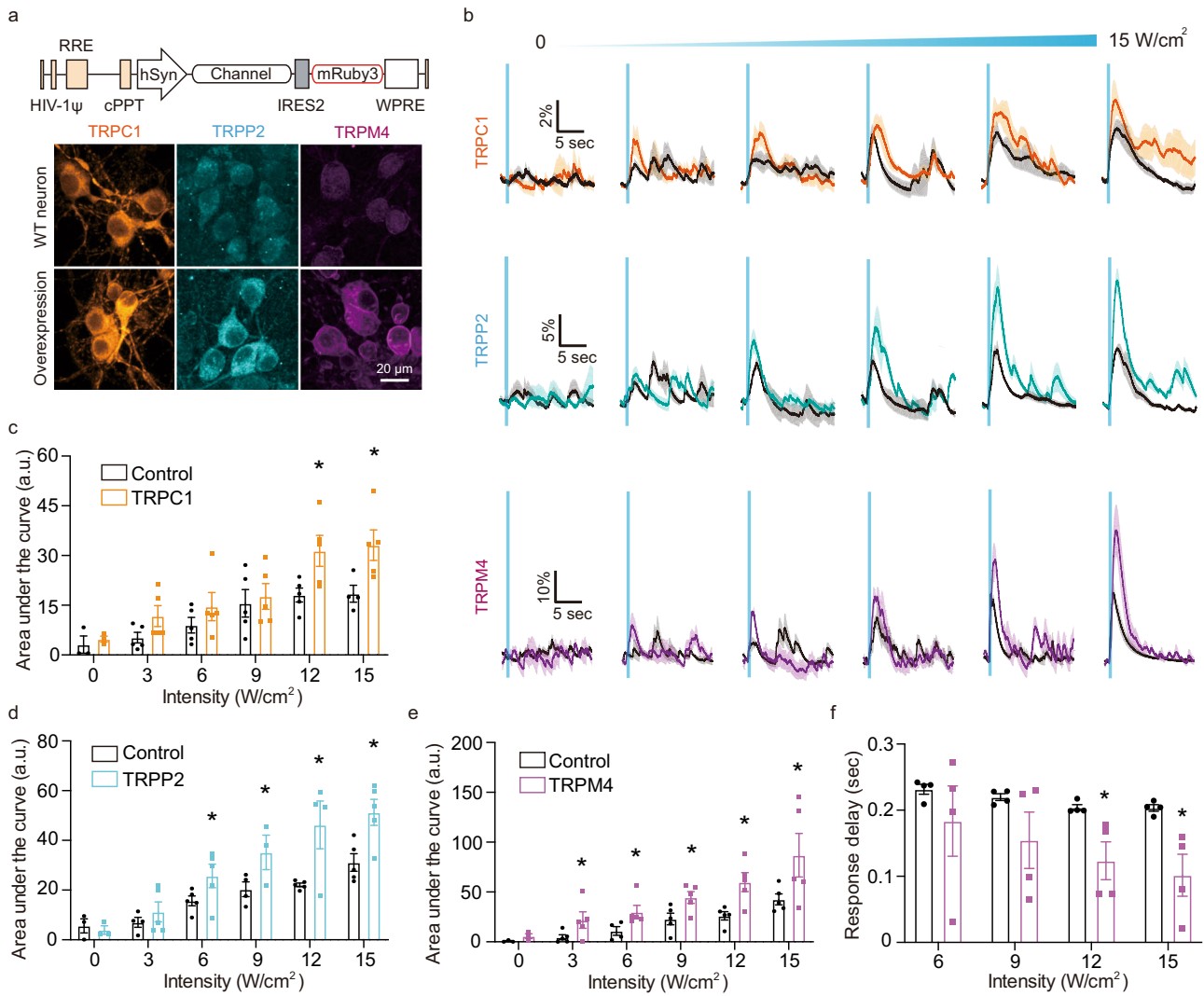

**Fig. 7 Neuronal response to ultrasound is enhanced by overexpression of mechanosensitive and amplifier channels. a** Schematic of genetic constructs for overexpressing TRPC1, TRPP2, and TRPM4 and representative immunostaining images for the channels with and without overexpression. **b** Calcium responses from wild type neurons and overexpressing neurons as function of ultrasound intensity and quantification of area under the curve for **c** TRPC1 ($n = 5$ independent experiments each, Unpaired $T$-test, two-tailed, $p = 0.0316$ (12 W/cm$^2$), $p = 0.0369$ (15 W/cm$^2$)), **d** TRPP2 ($n = 5$ independent experiments each, Unpaired $T$-test, two-tailed, $p = 0.0855$ (6 W/cm$^2$), $p = 0.0795$ (9 W/cm$^2$), $p = 0.0105$ (12 W/cm$^2$), $p = 0.0127$ (15 W/cm$^2$)), and **e** TRPM4 ($n = 5$ independent experiments each, Unpaired $t$-test, two-tailed, $p = 0.0815$ (3 W/cm$^2$), $p = 0.0578$ (6 W/cm$^2$), $p = 0.0317$ (9 W/cm$^2$), $p = 0.0114$ (12 W/cm$^2$), $p = 0.0841$ (15 W/cm$^2$)). **f** Comparison of response delay of calcium response between wild type and TRPM4-overexpressing neurons ($n = 5$ independent experiments each, Unpaired $T$-test, two-tailed, $p = 0.0321$ (12 W/cm$^2$), $p = 0.0442$ (15 W/cm$^2$)). Mean trace is solid and SEM is shaded. Bar graph values represent mean ± SEM.

excite cortical neurons. Ultrasound has a mechanical interaction with the cell, causing the opening of specific calcium-permeable mechanosensitive ion channels, including TRPP1/2, TRPC1, and Piezo1. Calcium ions accumulate at a relatively low level over approximately 200 ms until they trigger the opening of calcium-sensitive sodium channels, including TRPM4. This leads to depolarization of the cell membrane and the opening of voltage-gated calcium channels, including T-type channels, leading to the large responses observable by GCaMP6f imaging (Fig. 6a). While this signaling pathway is consistent with the response kinetics measured in our experiments, it is not possible to rule out contributions from more direct activation of voltage gated ion channels, which were described in previous work[25]. The latency of the observed response, which decreased with increasing stimulation intensity, is, however, consistent with the response latency observed across multiple animal studies (70–300 ms)[4,6,10,11,17].

The use of primary cortical neurons as a model system allowed us to dissect the mechanisms of ultrasonic neuromodulation in comprehensive detail in the absence of the potential artifacts confounding in vivo studies, such as indirect auditory excitation[30,31]. To ensure that our results are relevant for in vivo scenarios, we took care to culture neurons on an acoustically transparent substrate and confirmed that our ultrasound conditions elicited responses under both 2D and 3D culture conditions. Indeed, key features of the cultured neuron response to ultrasound matched those observed in vivo[4,6,10,11,17,45], including response latency and the range of responsive ultrasound intensities. Differences between our findings and previous studies examining the response to FUS in vitro may be due to specifics of how cells were stimulated. For example, cells cultured on hard plastic or glass substrates may experience different mechanical and acoustic conditions than cells stimulated on top of acoustically transparent materials or within soft gels.

The stimulation of neurons is repeatable, non-toxic and dose-dependent on ultrasound intensity and pulse duration. Our data rule out temperature and synaptic neurotransmission as essential mechanisms underlying ultrasonic neuromodulation. In addition, ultra-high frame rate imaging revealed no large-scale deformation or cavitation on the timescales of either the ultrasound cycle or the ultrasound pulse. Furthermore, no evidence was found to support a mechanism involving intramembrane cavitation and charge accumulation[24,29], which predicts the formation of bubbles within a large fraction of the cell membrane, as well as strong hyperpolarization during FUS application. We observed no dependence of the neuronal response on solution degassing, which is expected to affect bubble dynamics, and our ultra-high-speed camera also did not record major changes in the appearance of the membrane, as might be expected with extensive formation of bubbles (whose refractive index differs from water). Furthermore, our imaging of membrane potential during FUS application did not reveal hyperpolarization. However, the limitations of our experimental tools prevent us from conclusively ruling out the intramembrane cavitation hypothesis: the optical resolution of our setup (>500 nm) is insufficient to visualize very small bubbles, and our voltage indicator does not have sufficiently fast kinetics to observe voltage fluctuations on the ultrasound timescale.

Several questions remain open for further study. Our experiments suggest, via the roles found for the actin cytoskeleton and mechanosensitive ion channels, that ultrasonic neuromodulation is mediated by mechanical stress on the plasma membrane. However, the precise forces and nanoscale deformations caused by ultrasound remain a subject for future research, which could include multiscale computational modeling and biophysical techniques specifically designed to measure nanoscale motion[69,70]. These studies should further distinguish the roles of both traveling waves and static pressure gradients generated by beam focusing and reflections, which may be present in both in vivo and in vitro preparations[38]. In addition, while our study identified TRPP1/2, TRPC1, and Piezo1 as mechanosensitive ion channels involved the ultrasound response, the incomplete efficiency of our CRISPR knockouts makes it difficult for us to assess their relative roles. Alternative knockdown methods such as RNA interference or experiments with neurons derived from transgenic animals could provide further quantitative information. The relative expression of these channels in different neuronal sub-types may also impact the extent to which various populations of neurons in the brain respond to ultrasound. Indeed, recent studies are starting to examine the contributions of various cell types to the brain's response to FUS using intracranial electrical recordings[28,34,35,71]. These studies could be extended in the future by performing in vivo knockdowns of specific ion channels. Care must be taken to control for artifacts arising from the interaction of sound waves with recording electrodes and auditory side-effects[30,31], which are not present in our in vitro preparation.

Furthermore, the insights obtained in our study concerning the ion channels involved in ultrasonic neuromodulation may inform the development of sonogenetic strategies to sensitize specific brain regions and neuronal sub-populations to ultrasound[41,72]. Indeed, we showed that the overexpression of TRPC1, TRPP2, and TRPM4 increased the sensitivity of cortical neurons to ultrasound at reduced pulse intensities and durations, and in the case of TRPM4 greatly accelerated the response kinetics. Future work should focus on co-expressing these and other proteins identified in our study and applying them as sonogenetic agents in vivo.

In addition to cortical neurons, it would also be interesting in future studies to examine the biophysical and molecular bases of ultrasonic stimulation in other cell types. For example, recent studies have demonstrated ultrasound-enhanced cholinergic signaling in the spleen[73,74], insulin release from pancreatic beta cells[75] and bone fracture healing[76] in a calcium-dependent manner. Furthermore, overexpression of the mechanoreceptors and amplifier channels identified in this study could sensitize cells that do not have intrinsic ultrasound responses, which would be of interest for both further mechanistic study (e.g., in spiking HEK cells) and the development of sonogenetic tools. We anticipate that the mechanistic insights obtained in this study will help stimulate each of these future research directions.

## Methods

**Primary neuron preparation.** All animal procedures were approved by the Institutional Animal Care and Use Committee of the California Institute of Technology. Custom cell culture dishes were prepared from 3.5 cm diameter glass-bottom dishes (35 pi, Matsunami, Osaka, Japan). The inner glass was removed by a diamond tip scribe (Fisher Scientific) and Mylar thin film (Chemplex, 2.5 μm thickness) was attached to the bottom of dish by polydimethylsiloxane (PDMS, sylgard 184, Dow), then baked for 3 h at 40 °C. Surfaces of the Mylar film were coated by poly-D-lysine (0.1mg/ml in Trizma buffer, pH 8, Sigma) overnight, and washed with deionized water followed by 70% ethanol and dried. Cortical tissues were dissected from embryonic day 18 C57BL/6J mice (The Jackson Laboratory). The tissues were rinsed with Hank's Balanced Salt Solution (VWR) and dissociated by pipetting, followed by centrifugation at 1000 rpm (0.2 g) for 2 min. Pellet was collected and re-suspended in culture medium. Cells were seeded on the top of Mylar dish at a density of 100 cells/mm² (for ultrasound stimulation experiments with minimum spontaneous activity) or 300 cells/mm² (for measuring spontaneous activity), and maintained in Neurobasal medium (Thermo Fisher Scientific) supplemented with B27 (2% v/v, Thermo Fisher Scientific), GlutaMax (2 mM, Gibco), glutamate (12.5 μM, Sigma) and penicillin/streptomycin (1% v/v, Corning) in a humidified incubator with 5% CO₂ and 37 °C. BrainPhys neuronal medium supplemented with SM1 (STEMCELL) was used in experiments involving channel overexpression. Half of the medium was changed with the fresh medium without glutamate every 3 days, and neurons were used for ultrasound stimulation experiments after 12–14 days from the seeding.

For 3D neural tissue culture, Mylar dishes were pre-treated with oxygen plasma for 1 min. Fibrillar collagen (Collagen I, Rat Tail, Gibco) was diluted to 2 mg/ml to mimic the stiffness of intact brain[77]. The center of the dish was filled with 200 μl of the collagen mixture and incubated at room temperature for 30 min and washed with fresh medium. Re-suspended cells (50 k cells) were mixed with 100 μl of the collagen mixture and gently deposited onto the pre-gelled collagen, incubated for 30 min and washed with fresh medium. One milliliter of the collagen mixture was then added to the dish and incubated for 1 h (for a total thickness ~1 mm). Then 1 ml of fresh medium was filled after washing with the culture medium.

For calcium imaging, Syn-driven GCaMP6f as a calcium sensor was delivered to neurons via AAV1 viral vector transfection (Addgene 100837-AAV1, 1E10 vp/dish) at 4 days in vitro. Membrane potential was optically imaged using an Ace2N voltage sensor[78]. To construct the Ace2N-4AA-mNeon voltage sensor, the first 228 residues of the Acetabularia acetabulum rhodopsin II protein (GenBank: AEF12207) were codon-optimized for mouse cell expression and the cDNA was synthesized commercially (Integrated DNA Technologies). This was fused using a 5-residue linker (MLRSL) to the mNeonGreen protein (residues 14–236, GenBank: AGG56535), which was fused directly to a Golgi trafficking sequence (KSRITSEGEYIPLDQIDINV) and ER export tag (FCYENEV). The construct was cloned into a lentiviral transfer vector containing the woodchuck hepatitis virus posttranscriptional regulatory element (WPRE) (pLVX series, Clontech, Mountain View, CA) under the human synapsin 1 promoter (hSyn) with a strong Kozak sequence (GCCACC) using Gibson assembly. Lentiviral packaging was performed in HEK 293T cells using commercial plasmids (Addgene plasmids 12259 and 12263) and protocols. Lentivirus was applied to neurons at 3 days in vitro (1E9 vp/dish).

In preparation for voltage imaging and ultrasound stimulation under calcium-free conditions, the culture medium was replaced with artificial cerebrospinal fluid (ACSF) containing (in mM) 25 NaHCO₃, 10 D-glucose, 125 NaCl, 2.5 KCl, 1.25 NaH₂PO₄, 1 MgCl₂6H₂O, 2 CaCl₂2H₂O (0 CaCl₂2H₂O add 1 EGTA for calcium free ACSF) equilibrated with 5% CO₂. After the media replacement, cells were allowed to recover for 30 min in incubator.

For degassing the medium, 25 ml of fresh medium in a 50 ml tube was placed in a vacuum chamber to apply negative pressure (Welch, IL, −0.1 MPa). Boiling of the medium was seen in the first 5 min, and additional degassing for 55 min was performed. After the degassing, normal culture medium was replaced with the degassed medium, and cell were allowed to recover for 30 min in incubator. The diffusion time of O₂ or CO₂ (>12 h for the 1 cm diffusion depth)[79] was much slower than the total experiment time (45 min).

**Ultrasound stimulation setup and characterization of transducer.** A 300 kHz ultrasound transducer (BII-7654/300IM, Benthowave Instrument INC. Canada) with 50 mm diameter and 24 mm focal distance was used in all experiments where

300 kHz ultrasound was applied. The transducer was submerged in degassed water (degassed by a water conditioner, Onda, Aquas-10) and angled 20° relative to normal incidence for the Mylar film using a customized holder. An Axon Digidata 1550 acquisition system (Molecular Devices, CA) was used to program and generate a set number of trigger pulses that were sent to an arbitrary waveform generator (Tabor Electronics, WX1282C) to generate the desired number of cycles of a sine wave at 300 kHz. The output of the generator was amplified by a linear amplifier (75A250A, RF Microwave Instrumentation, PA) and used to drive the transducer. Calibration of the transducer and measurement of the pressure profile were done using a fiber optic hydrophone system (FOH, Precision Acoustics, UK) and optic hydrophones (PFS and TFS, Precision Acoustics, UK). The position of the hydrophone was controlled by stepping motor controllers (VELMEX INC., NY) while voltage traces were recorded by a digital oscilloscope (DSOX2004A, Keysight, CA) connected to a PC. From these measurements, the acoustic intensity of the ultrasound stimulus waveforms was calculated based on published standards[2]. To characterize the neuronal response to ultrasound at different acoustic intensities and durations, we randomized the sequence of the different waveforms to avoid accumulation effects. For neural stimulation with 670 kHz frequency, a 670 kHz ultrasound transducer (TXH-0.67-75, Precision Acoustics, UK) was used with the same configuration as described above.

### Fluorescence imaging of calcium and voltage.

A 490 nm LED light (LED4D067, Thorlabs, NJ) was used to excite the fluorescent proteins, and emitted signals were collected by an immersion lens (10×, NA 0.3, Leica) and recorded by a sCMOS camera (Zyla 5.5, Andor) at 100 Hz (200 Hz for voltage imaging). The recorded images were processed to extract calcium or voltage signals (d$F/F$) from each neuron by using NeuroCa[80]. Fifty to three hundred cell bodies per each ROI (dish), depending on their seeding density, were detected by NeuroCa. Single-cell calcium signals from the ROI (dish) were averaged, and the averaged signals from each dish were used for data plotting. The number of biological replicates ($n=$) and ± SEM were based on the number of dishes. Intensities during 500 ms before the onset of ultrasound stimulation were averaged, and this average was used as a baseline to calculate the area under the curve response to stimulation. Calcium signals within a time window between 0 s (onset of ultrasound) to 5 s were used for the calculation of area under the curve. To calculate calcium response delay, calcium signal was fitted using sigmoid fitting method (4-parameter logistic regression, R-square >0.95), then 0.2% increase in the magnitude was set as onset time. Variations in overall response amplitude arise due to the use of different neuronal preps on different experimental days, which was required to test this large number of experimental conditions. Neurons prepared on different days (obtained from a unique animal and transduced to express GCaMP) exhibited some variation in calcium responses. For each test condition, we therefore included a matching control from the same neuronal preparation. In some cases, we were able to run multiple conditions together on the same experimental day, in which case they shared a control. To perform voltage imaging in a generic model of an excitable cell, we used spiking HEK cells (a gift from Adam E. Cohen) which was cultured as previously described[81]. We cloned an EF1a-Ace2N-mNeon construct into a lentiviral transfer vector and performed lentiviral packaging using the protocol described above. Lentivirus (1E10 vp/dish) was applied to the spiking HEK cells at 50% confluency and centrifuged down onto the cells at 1500×$g$ for 90 min with 10 µg/ml polybrene. After 3 days of incubation at 37 °C, cells were treated with trypsin-EDTA (0.25%, Gibco) for 1 min and plated on a mylar film dish at 80% confluency. After 24 h incubation at 37 °C, the cell medium was replaced with ACSF and incubated again for 30 min before voltage imaging. For voltage imaging of neurons, we used a higher recording speed (200 Hz) to observe the response latency. With this speed (and correspondingly reduced SNR), we were able to capture average firing responses from multiple dishes, while it was not able to picked up the distinct spikes from individual cells. All data were analyzed using custom code written in MATLAB (Mathworks, MA). All values represent mean ± SEM. Plots were generated using MATLAB and Prism.

### Ultra-high-frame-rate optical imaging.

To observe cell membrane deformation at MHz frequencies we used a Shimadzu HPV-X2 camera. Samples were illuminated using a 2W 532 nm laser (CNI, MLL-F-532-2W) controlled by an optical beam shutter (Thorlabs SH05, KSC101). Right-angle prism mirrors directed the laser light through a water bath and into a sample dish containing the imaged neurons. The transducer was positioned in the water tank at an angle of 45° relative to the water surface to minimize standing waves. 10× and 40× water immersion objectives (Leica, NA 0.3, Olympus, NA 0.8) were used. A series of prism mirrors and converging lenses with focal lengths of 200 mm and 50 mm delivered the image into the camera, which acquired 256 images over 51.2 µs. Images were acquired starting 100 ms after the onset of ultrasound stimulation, to capture events coincident with the initiation of calcium and voltage signals. As a positive control for detecting large scale deformation, a PDL-coated mylar film dish was biotinylated by incubating NHS-biotin (Thermo Fisher Scientific, 200 4 µg/ml) for 3 h. After washing the free linkers with PBS, the dish was then inverted and incubated for 1 h with streptavidin-functionalized microbubbles (Advanced Microbubbles Laboratories LLC SIMB3-4SA, 4 µm in diameter) to attach the bubbles onto the mylar film, and bubble cavitation was imaged using the same parameters as used with neurons.

### Cell viability and immunostaining.

Primary neurons were pre-treated with Calcein AM (Thermo Fisher Scientific), and live cells were imaged using 490 nm fluorescent excitation. Then, the neurons were stimulated with the highest intensity and longest duration ultrasound (15 W/cm², 500 ms, 30 times with 20 s inter-pulse interval) and imaged again after a 1 h incubation. Live and dead cell counting was performed using ImageJ (NIH) to calculate the cell viability[82]. For immunostaining, primary neurons were fixed using ice-cold paraformaldehyde (4% in PBS, VWR) for 10 min at 4 °C, and washed with PBS. Nonspecific biding was blocked by 6% bovine serum albumin (Sigma) for 30 min at room temperature and cells were washed in PBS. Primary antibody (anti beta-tubulin (1:500, Sigma), Alexa Fluor 488 Phalloidin (1:500, Thermo Fisher Scientific), anti-TRPC1 (1:200, Alomone Labs), anti-TRPM4 (1:200, Alomone Labs) and anti-TRPP2 (1:200, Alomone Labs)) were diluted in 1.5% bovine serum albumin, and incubated with cells for 1 h. After washing with PBS for 3 times, secondary antibodies (Alexa Fluor 488 or 594 or 647 (1:200,Invitrogen)) that were diluted in 1.5% BSA were loaded to neurons for 1 h at 37 °C. After washing with PBS, Hoechst 33342 (1:200, Sigma) was added to the PBS solution for nuclear staining. After 10 min, cells were washed with PBS, and imaged using a confocal microscope (LSM 880 with Airy scan, Zeiss).

### Pharmacological treatments.

Chemical blockers or peptide inhibitors (all from Tocris Bioscience, NM) applied directly in the media were used to block ion channels or manipulate cellular pathways. Identical volumes of buffer solutions were applied to control samples. A minimum concentration of TTX (final conc.: 1 µM) was titrated by monitoring the change of spontaneous calcium activity, and this was used to pharmacologically block voltage-gated sodium channels. Thapsigargin (TG, final conc.: 500 nM) was used to block calcium pumps in the ER. Calcium release from ER after the TG application was confirmed by a transient calcium signal increase (Supplementary Fig. 7)[83]. To block the presynaptic inputs, the postsynaptic blockers AP5 (final conc.: 1 µM) and CNQX (final conc.: 1 µM)[84] were used. Actin filaments were depolymerized by their specific inhibitors, cytochalasin D[49] and vinblastine, respectively[85]. Spontaneous calcium activities from separated groups were recorded before and after 1 h from the inhibitors treatment (final conc.: 1 µM), and the neurons were stimulated by ultrasound. After finishing the stimulation experiments, neurons were fixed for immunostaining. Minimum and working concentrations of chemical or peptide channel blockers were investigated by measuring the change of spontaneous calcium activities before and after applications (Supplementary Fig. 8). Gadolinium[55] was applied to nonspecifically block the mechanosensitive ion channels (final conc.: 20 µM). After ultrasound stimulation, neurons were washed by fresh medium and incubated for 30 min for cell recovery, followed by ultrasound stimulation. Ruthenium red (final conc.: 1 µM)[57] and TTA-P2 (final conc.: 3 µM)[67] were used before ultrasound stimulation to block TRP channels (TRPV1, 2, 4) and T-type calcium channels, respectively. To inhibit GPCRs, suramin[58] was added to medium (final conc.: 60 µM) and incubated with cells for 1h, then stimulated cells with ultrasound. GsMTx4[59] was added to medium (final conc.: 10 µM) and incubated with cells for 2h to inhibit Piezo1 and TRPC1 channel gating, then stimulated cells with ultrasound.

### CRISPR/Cas9 for ion channel knockout.

Three sgRNAs for each target channel were designed using CRISPRko[61]. Each sgRNA was inserted into a LentiCRISPR-mCherry backbone (Addgene, #99154) and cloned by an established protocol[86]. Lentivirus containing the sgRNA was delivered to neurons (1E9 vp/sample) at 3 days in vitro. After 10 days, genomic DNA from the neurons was extracted using a DNA extraction kit (Qiagen), and CRISPR target regions were amplified by PCR. The PCR products were sequenced (Sanger sequencing), and the sequencing results were compared with those from wild-type neurons and non-targeting sgRNA to confirm the CRISPR knockout and to estimate knockout efficiency using the Tide tool[63]. The most effective sgRNA was then selected from among the three sgRNAs (Supplementary Table 1) and its non-specific targeting efficiency was estimated by CFD score[61] (Supplementary Table 2).

### Gene overexpression.

The mouse TRPV1 (GenBank: AB040873.1), TRPP2 (Gen-Bank: BC053058) and TRPM4 (GenBank: BC096475), human TRPC1 (GenBank: Z73903.1), genes were synthesized commercially (Integrated DNA Technologies) and cloned upstream of an internal ribosome entry site (IRES2) and mScarlet (TRPC1, TRPP2) or mRuby3 (TRPV1, TRPM4) gene. The construct was inserted into the same lenti-backbone as described above. The viral particles were added to neurons at 3 days in vitro (1E9 vp/sample) and maintained for 10 days. To measure temperature change during ultrasound stimulation using mCherry, hSyn-driven mCherry was inserted into the lenti-backbone by Gibson assembly. The viral particles were added to neurons at 3 days in vitro (1E9 vp/sample), whole media was replaced with the fresh media at 4 days in vitro, and the cells were maintained for 6 additional days.

### Reporting summary.

Further information on research design is available in the Nature Research Reporting Summary linked to this article.

## Data availability

Genetic constructs will be made available through Addgene. Source data are provided with this paper.

## Code availability

Processing code is available upon request to the authors.

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

## Acknowledgements

The authors thank Minjee Jang for assistance with calcium image processing and helpful discussions, Tomokazu Sato for assistance with initial experiments, and all members of the Shapiro lab for helpful discussions and assistance with experiments. This research was supported by NIH BRAIN Initiative grants R24MH106107, RF1MH117080 (to M.G.S.) and NARSAD Young Investigator Grant (28802) from the Brain & Behavior Research Foundation. Related research in the Shapiro Lab is supported by the Packard Fellowship.

## Author contributions

S.J.Y. and M.G.S. conceived this research. S.J.Y. and M.G.S. designed all experiments and S.J.Y. performed the experiments and analyzed the data. D.R.M. and S.J.Y. planned and performed the high-speed-imaging. R.C.H. and S.J.Y. designed and built the genetic constructs. J.L. designed the initial optical setup for calcium imaging. S.J.Y. and M.G.S. wrote the manuscript with input from all authors.

## Competing interests

The authors declare no competing interests.

## Additional information

**Peer review information** *Nature Communications* thanks Jacob Robinson and the other anonymous reviewer(s) for their contribution to the peer review this work. Peer reviewer reports are available.

