## [Peer Review File · Nature Communications]

Reviewers' Comments:

Reviewer #1:

Remarks to the Author:

Yoo et al. conducted and described a comprehensive study to elucidate the mechanistic nature of ultrasound neuromodulation effects on primary cortical neurons. By ruling out the heating/cavitation/deformation factors, a mechanical mechanism of ultrasound-neuron interaction through a specific signaling pathway and ion channels was identified through extensive in vitro tests on GCaMP6f-expressing neurons and quantifications based on epifluorescence imaging. The overexpression of specific ion channels responsive to ultrasound neuromodulation through genetic manipulations further validated the significant contributions of TRPP1/2 and TRPC1 to the ultrasound-evoked calcium response. Overall, this manuscript is well-written and would have broad interest and impact. Specifically, it has a broader scope by including more thorough control studies using pharmacological treatments and gene manipulation to narrow down the biophysical basis of ultrasound neuromodulation than the pioneer work did in the Reference 25 by Tyler et al.; however, comparing these two works, the authors may need to further justify the following two fundamental but different observations from the calcium transients.

1. Minimal time delay between the onsets of ultrasound stimulation and the calcium response was reported in the Ref. 25, but in this manuscript, the calcium signal showed an "approximately 200 ms" delay to achieve a thresholding, i.e., 0.2% signal magnitude increase. Based on this delay, the authors believed that it "reflects the kinetics of the neurons' response to FUS". This can be explained by the proposed signaling pathway, in which the calcium mediates ion-channel cascading effects. However, this cascading pathway seems not including the more direct effects of ultrasound on ion channels as reported by the Ref. 25 that the low-intensity and low-frequency focused ultrasound is able to activate voltage-gated sodium channels and stimulate voltage-dependent calcium transients. Please discuss. In addition, why were the high-speed optical images "acquired starting 100 ms after the onset of ultrasound stimulation" instead of aligning the image acquisition with the ultrasound event?
2. In Ref. 25, the focused ultrasound was observed to trigger synaptic vesicle exocytosis and synaptic transmission, but in this manuscript, the authors suggested that "each neuron responses to ultrasound on its own" based on the non-significant change of calcium response with the synaptic blockers being administered. Although the difference between the control and treatment groups was not statistically significant, the decreased mean value is still obvious. Also, the limited sample size in Fig. 2j may also lead to the non-significant observation.

Other major issues need to be addressed are:

1. The comparison between 300 kHz and 670 kHz is interesting. The authors "found that no significances in response amplitude or onset delay" in line 49 on page 4. However, in the caption of Supplementary Fig. 1g, the p value is described as 0.0011. Why is this comparison considered as not significant? What is the alpha level being used here?
2. A chirped waveform was used, and the authors mentioned in line 60 on page 4 that "we found the calcium signal unaffected in terms of response amplitude" as the data shown in Supplementary Fig. 4. The AUC may not be affected from the panel b, but it can be observed that the response onset delay is different. Is this difference statistically significant?
3. In lines 88-89 on page 6, the authors mentioned that "the acoustic radiation force on the neurons due to FUS is expected to be weak". Any evidence to justify this claim?
4. In Fig. 4d and Supplementary Fig. 8b, the calcium responses, including the spontaneous activities, did not change significantly before and after blocking TRPV1, 2 and 4 channels using RR. As mentioned in the Methods, "minimum and working concentrations of chemical or peptide channel blockers were investigated by measuring the change of spontaneous calcium activities before and after applications", how did the authors determine the minimum 1 μ M for RR? I wonder whether an increased RR concentration would lead to a significant change of calcium transients.
5. Based on these in vitro experimental observations, it seems that the intramembrane cavitation theory and the NICE model are not valid/accurate to explain the ultrasound neuromodulation. Please further discuss this topic.
6. Please discuss if the in vitro findings could be extended to in vivo systems. It would be nice if the authors could discuss this in light of recent in vivo reports.

Minor issues and typos need to be clarified and fixed:

1. In Fig. 1b, what does the step waveform (above the raw ultrasound waveform) stand for?
2. In line 44 on page 4, the figure reference should be "Supplementary Fig. 1b".
3. In the caption of Fig. 5h, please remove the redundant Δ .
4. In the lines 184-185 on page 12, the reference to Supplementary Fig. 7e seems incorrect here. Please check.
5. In the line 203 on page 13, where is Fig. 7, h-j?
6. In line 383 on page 19, change "...were treated..." to "...were used ...".
7. The supplementary tables shared the same number and title. Please correct this.
8. The last column of Supplementary Table 2 should be "CFD score" instead of "CDF score".

Reviewer #2:
Remarks to the Author:

A. Summary of the key results

In this manuscript, Yoo et al. dissect the mechanism of action by which focused ultrasound (FUS) excites neurons, through calcium influx and depolarization of the cell membrane. Their approach is based on calcium and voltage fluorescent imaging, combined to specific pharmacological and genetic inactivation of key candidate channels that might be involved in mediating the cellular response to FUS. The main parameters reported are the intensity and the delay of the calcium response upon FUS stimulation, and how they are impacted by channel modulations.

This elegant and effective approach manages to draw a convincing model for the excitation of cells after stimulation by FUS, where the mechanical stimulation of mechanosensitive calcium channels leads to the potentiation of sodium channels (such as TRPM4) that amplify the response and initiate a stronger, but delayed, calcium influx through voltage-gated calcium channels. This model is consistent with the response delay reported and provide major inputs on how to improve the kinetics of sonogenetics.

Overall, this manuscript is well written, the data is presented clearly, and the claims well justified by the careful experiments performed by the authors. Moreover, this kind of mechanistic work is incredibly valuable to the community who works on understanding new neuromodulation tools to create more effective techniques. This work will be an important foundational study for future efforts. It is a rare treat to encounter a manuscript as well-crafted as this one where it is difficult to find any major weaknesses. I want to congratulate the authors on this work, and I do not believe any additional experiments are required for publication. Nevertheless, I would be a poor reviewer if I did not offer suggestions for improvement. I believe these issues can be corrected by minor updates to the text. I do NOT think additional experiments are necessary at this point. These critiques are intended only to help strengthen and clarify this excellent work. Note I prepared these with significant help from a research scientist in the lab and we both agree on these critiques:

Primary concerns:

- We don't think a local increase in temperature at the membrane level can be entirely ruled out as a contributing factor based on the data presented here. Measurements done by optical thermometer measure bulk changes. Using RFP as a thermal reporter shows cytoplasmic changes with a sensitivity of 0.1 fluorescence per 5 °C, and only measures intracellular bulk temperature, not any local temperature on the membrane. It would be reasonable to assume that at the relevant timescales the temperature would be uniform throughout the cell, but this assumption should be specified in the text and justified if possible.
- Similarly, membrane cavitation may not be entirely excluded as a contributing factor based on these data. Is it possible that the membrane can move to the extent that it may not be clearly visible from the microscopy data? What is the sensitivity of this experiment? Could you say that you can exclude cavitation larger than a specific displacement value? This should be made clear in the manuscript (results and discussion)
- We would like to see a discussion on why there remains a 40% response in the presence of Gadolinium which should inhibit all mechanical response.
- Over-all, we think the response delay should be quantified throughout the text, for different conditions and different recording (calcium and voltage measurements) so that we can compare if the perturbations affect the delay.
- Discuss/explain: why do control recording have such different amplitudes (scales are different). In the CRISPR figure, specify what the control is (we are assuming it is a non-targeting sgRNA). Why are there variations in the response amplitude of non-targeting sgRNA between experiments? Why are the same controls used at times (ex: Fig. 5 d and e, or f and g). We are sure there are good reasons for why the controls are repeated in some experiments and not others, but this should be clarified for the reader
- Most traces: why show the sem rather than the individual traces to give a better sense of traces variability? Especially concerning delay. Perhaps a supplemental figure would be good here.
- We would be curious if FUS stimulation could be completely inhibited (other than by removing extracellular calcium), for example through multiple inhibition (drug cocktails). + Is there a dose dependence effect of major inhibitory drugs? Alternatively, since FUS doesn't work well on spiking HEKs, could response be improved by expressing missing channels that are essential in neurons' response to FUS?

B. Originality and significance

Although mechanosensitive channels have previously been proposed as a major effector in FUS stimulation of neurons (e.g., Tyler et al 2008, Kubanek et al. 2016, Prieto 2018), this paper carefully demonstrate their roles, and dissects the mechanism by which the initial calcium influx is amplified to trigger depolarization and subsequent stronger calcium influx and provide an explanation to the response delay. Therefore, these results have a significant impact on the research field, by indicating which pathways and proteins to engineer to effectively decrease the response time of FUS stimulation. Improving the response delay is essential for future FUS applications in neuromodulation.

C. Data & methodology: validity of approach, quality of data, quality of presentation

The experimental design based on fluorescence imaging is simple and effective, allowing for repeated recordings for each condition tested here, including controls. Consequently, the statistical analysis is strong, and the conclusions are convincing. The data are well presented, and the different experimental conditions are well described with schematics in figures, which help the overall comprehension of the paper.

The only minor concerns I have about data presentation is that I would prefer to see the individual calcium and voltage traces rather than the average +/-sem, in order to see the variation in amplitude and response delay from experiment to experiment for each condition. Also, it is not always clear what the control trace is for each condition, and why the amplitude of the response (including of the control) varies so much between experiments. I think this should be discussed in the manuscript.

I also think that the data for response delay should be presented for every condition where the response amplitude is shown. This parameter is equally important and should be shown whenever possible. Importantly, it should definitely be analyzed and plotted for the voltage imaging, in order to compare the calcium delay and the voltage delay (we don't expect any significant difference based on the mechanism of amplification proposed here).

D. Appropriate use of statistics and treatment of uncertainties

T-test and Anova test are used appropriately when needed (except missing in figure 2b). However, it is not clear how the sem are calculated, are the authors using n as the number of dishes recorded, or the number of cells recorded (the grayed area indicating the SEM is often very small, hence my question). This should be clarified in the legends.

E. Conclusions: robustness, validity, reliability

The conclusions concerning the amplification of the signal, and the effect on the response delay are well presented, convincing and based on the interpretation of the data.

However, I would be a little more careful in dismissing a temperature or cavitation effect based on the experimental data presented here. Could membrane cavitation occur without changing the appearance of the membrane, and therefore be undetected? Cavitation is not my field of expertise, so I would leave it to other reviewers' evaluation.

As for temperature, there could be a temperature increase at the membrane level, which would not be detected by optic measurement, or by using mRuby. (There might be a way to refine temperature measurement, maybe by measuring membrane diffusion (FRAP) or by targeting mRuby to the membrane). This does not question the validity of the conclusions of this study concerning which channels are involved in the mechanism of action and how the signal is amplified. But even under mechanisensitive channel inhibition with Gadolinium, the response is 40%, hinting that there might be an additional "entry point". Moreover, mechanosensitive channels can be modulated by local temperature changes. This could also be solved by simply sating the timescale over which you expect to reach thermal equilibrium and how that compares to the channel dynamics – it's probably safe to say you're in thermal equilibrium

I don't think more experiments are needed, but I am not sure we can completely "rule out" temperature as an actor in the initiation of the response. I would suggest more careful phrasing, and make sure that the highlight of this paper is the amplification of the signal, which provides a way to improve the kinetic of FUS, rather than the certainty that the original stimulation is through mechanical perturbation.

F. Suggested improvements: experiments, data for possible revision

The data presented here represent a lot of well performed experimental work, and the data provide a strong model for the amplification and response delay of FUS stimulations of neurons.

I would suggest a few improvements in the phrasing, and in the data presentation, as discussed above.

- rephrasing the "ruling out" of temperature, and cavitation,
- showing all individual fluorescent traces when possible (instead of average +/- sem). Some responses seem to start before the stimulus, and showing all traces would clarify how often that happens (ex, Fig. 4d)
- I think response delay is as important of a parameter as response amplitude, and it should be measured and plotted when possible (in Fig. 2, Fig. 3, Fig. 4, Fig. 5, and Fig. 6; also in SF 4)

- For traces and area under the curve, for each FOV (for each dish) it is not clear if the fluorescence of each cell is measured, then averaged, or if the fluorescence of the entire FOV is extracted. Please clarify in methods or in legends.

I do not require any supplemental experiment prior publication, unless the authors want to strongly dismiss temperature as possibly playing a part. In this case, I would suggest a more precise reporting of temperature changes / perturbations at the cell membrane level during FUS (indirect measurements could be: using a thermally-sensitive channel in an otherwise unresponsive cell line, measuring changes in membrane fluidity (FRAP))

G. References: appropriate credit to previous work?

To my knowledge, previous work is properly cited.

H. Clarity and context

No concerns. Well written, and clear figures.

Figures

Fig. 1: How is the SEM calculated? I am assuming with $n=4$ dishes, but clarify (Material and Methods mentions 50-300 cell bodies (!)). For each FOV, it is not clear if the fluorescence of each cell is measured, and an average trace is calculated for this FOV, or if the fluorescence of the entire FOV is extracted at once.

Fig. 2: Add statistical analysis for *2b* (temperature increase). *2c* needs to show more traces and statistical analysis. *2g* could bubble form earlier than 100 msec after initiating FUS, and have a delayed effect on calcium? *2i* The response for Cyto.D seems to start before the stimulation. Is there a sliding average problem? Or an outlier trace (which would be made clear by plotting all the fluorescence traces).

Fig. 3: report response delay time measurement for each recording with stats. Also, *3d* why not show the different traces in the inset (they are visible on the main trace).

Fig. 4: Why are the traces so variable in intensity (see scale bars)?

Fig. 5: same question as Fig. 4.

Fig. 6: all good

Fig. 7: these are great experiments. I think the Response delay data for TRPC1 and TRPP2, even if not significant, should be shown as supplementary plots.

Supplemental Figures:

Sup. Fig. 3: I think data should be from $n=3$ dishes since cells are so confluent. (calcium will propagate between cells, making the $n= x$ cells irrelevant). Legend should be more precise on stimuli "15 W/cm², 500 ms, 30 times with 20 sec inter-pulse interval"

Sup. Fig. 3 & Sup. Fig. 4: the 3D responses as well as chirp waveform stimulation seem more delayed (start at the end of the stimuli). Having a plot of the response delay time would really help compare the

effect of all the conditions tested on timing. Here, specifically: why would chirp stim delay the calcium onset (or is it not significant)?

Sup. Fig. 5: a: is there a bar missing (there should be 6 conditions?). Also, intensities are not indicated for each frequency. Finally, PD and IPI are not explained.

Sup. Fig. 6: d: is it an average trace or a significant example? Could you show multiple examples. It would be interesting to see how often this small response to FUS occurs in spiking HEK.

Supplementary Tables both have the same number and title

Miscellaneous:

- Choose ms or msec
- Line 373-374: Thapsigardin data are not in supplementary figure 3. They are in supplementary figure 7.
- Line 375: missing tubulin "Actin filaments and tubulin were..."
- Line 380: change supplementary figure 7 to supplementary figure 8
- Line 385: is the suramin concentration, correct? (60 uM sounds like a lot)

JTR

Reviewer #3:

Remarks to the Author:

This manuscript under review describes an in depth study into the biophysical and molecular mechanism of ultrasound neuromodulation in an in vitro primary neuron system, and proposes a plausible hypothesis of mechanosensitive channel activation and calcium amplification. Ultrasound neuromodulation has been an emerging field with great clinical potential, yet the detailed mechanism remains elusive. This manuscript provides important new insight to the existing literature. The study itself is very thorough and clearly laid out, including dose and frequency dependency of ultrasound modulation, pharmacological inhibition of multiple mechanosensitive ion channels and the results are supported by genetic modifications. In addition, other hypothesis such as temperature and intramembrane cavitation has been carefully ruled out. Given the strong promise of this work, I suggest publication after revisions.

The average power of acoustic wave used in this study was 15W/cm², which seems to be higher than that used in other studies (e.g. 0.228W/cm² in <https://doi.org/10.1016/j.neuron.2010.05.008>).

The activation latency of 200 ms described in this manuscript seems to contradict previous reports of US induced EMG response with delay of less than 100ms.(<https://doi.org/10.1038/s41467-021-22743-7>, <https://doi.org/10.1016/j.brs.2019.03.005>) What are the possible factors contributing to this delay?

In fig3d, the voltage trace of the neurons do not resemble firing of action potentials as shown in fig S6b. Is this a subthreshold response? Are there trials where action potentials were observed after initial depolarization?

In fig 4a, in addition to block mechanosensitive ion channels, Gd will also alter all membrane mechanical properties by increasing membrane rigidity so that other possible mechanisms, if any, will also be suppressed.

In Fig 5cde, why do the calcium traces go below the baseline after stimulation?

We are grateful to all three reviewers for their enthusiastic evaluation of this work and helpful comments, which helped improve the manuscript. Referee comments are addressed in detail below. Changes in the manuscript are highlighted in blue font.

Reviewer #1 (Remarks to the Author):

Yoo et al. conducted and described a comprehensive study to elucidate the mechanistic nature of ultrasound neuromodulation effects on primary cortical neurons. By ruling out the heating/cavitation/deformation factors, a mechanical mechanism of ultrasound-neuron interaction through a specific signaling pathway and ion channels was identified through extensive in vitro tests on GCaMP6f-expressing neurons and quantifications based on epifluorescence imaging. The overexpression of specific ion channels responsive to ultrasound neuromodulation through genetic manipulations further validated the significant contributions of TRPP1/2 and TRPC1 to the ultrasound-evoked calcium response. Overall, this manuscript is well-written and would have broad interest and impact. Specifically, it has a broader scope by including more thorough control studies using pharmacological treatments and gene manipulation to narrow down the biophysical basis of ultrasound neuromodulation than the pioneer work did in the Reference 25 by Tyler et al.; however, comparing these two works, the authors may need to further justify the following two fundamental but different observations from the calcium transients.

→ Thank you for your positive review and helpful suggestions. We revised the manuscript to improve its connection to previous studies, and comprehensively addressed all other reviewer comments, as detailed below.

1. Minimal time delay between the onsets of ultrasound stimulation and the calcium response was reported in the Ref. 25, but in this manuscript, the calcium signal showed an “approximately 200 ms” delay to achieve a thresholding, i.e., 0.2% signal magnitude increase. Based on this delay, the authors believed that it “reflects the kinetics of the neurons’ response to FUS”. This can be explained by the proposed signaling pathway, in which the calcium mediates ion-channel cascading effects. However, this cascading pathway seems not including the more direct effects of ultrasound on ion channels as reported by the Ref. 25 that the low-intensity and low-frequency focused ultrasound is able to activate voltage-gated sodium channels and stimulate voltage-dependent calcium transients. Please discuss.

→ Thank you for pointing this out as an important topic for discussion. We added text on line 231 to address this question and point the reader more prominently to the previous study.

Line 231: While this signaling pathway is consistent with the response kinetics measured in our experiments, it is not possible to rule out contributions from more direct activation of voltage gated ion channels, which were described in previous work²⁵. The latency of the observed response, which decreased with increasing stimulation intensity, is, however, consistent with the response latency observed across multiple animal studies (70 ~ 300 ms)^{4, 6, 10, 11, 17}.

In addition, why were the high-speed optical images “acquired starting 100 ms after the onset of ultrasound stimulation” instead of aligning the image acquisition with the ultrasound event?

→ Thank you for this question. Because this ultrafast camera can only record 256 frames (51.2 μ s) of video, we had to choose a small recording window during the stimulation. In addition, we could only borrow this instrument (from our Aeronautics colleagues) for a very short time, allowing us to run just a few experiments. As a result, we could only record during one time window over several replicates. We chose to record at 100 ms because (1) our data suggested that this amount of time is needed to approach a measurable neuronal response in our system, (2) this timescale is sufficiently long to allow the cells to reach maximal deformation upon acoustic radiation force application, and (3) the bilayer sonophore model predicts that intramembrane bubbles could grow through multiple acoustic cycles until the appearance of the first action potential (predicted latency also \sim 100 ms). Imaging starting at 100 ms thus maximized our opportunity to see these phenomena. We updated the manuscript on lines 83, 97 to provide this rationale and state its limitations.

Line 83: Images were recorded starting 100 ms after the onset of FUS, providing sufficient time for bubble growth ²⁴ and approaching the latency of our observed neuronal excitation.

Line 97: While we did not perform ultra-high-speed imaging in the FUS application period before 100 ms, any deformation due to acoustic radiation force is expected to increase and persist on this timescale.

2. In Ref. 25, the focused ultrasound was observed to trigger synaptic vesicle exocytosis and synaptic transmission, but in this manuscript, the authors suggested that “each neuron responds to ultrasound on its own” based on the non-significant change of calcium response with the synaptic blockers being administered. Although the difference between the control and treatment groups was not statistically significant, the decreased mean value is still obvious. Also, the limited sample size in Fig. 2j may also lead to the non-significant observation.

→ Thank you for this question. We updated the text to state that a role for synaptic transmission cannot be formally ruled out as there may be a small effect not reaching statistical significance. At the same time, we can conclude that synaptic transmission is not *necessary* for neurons to respond to FUS, since blocking it did not strongly abrogate responses. We note that similar sample sizes were more than sufficient to statistically detect effects of numerous other perturbations in this study. We also added text to the discussion pointing out differences between our study and others that may have led to divergent results.

*Line 107: After treating the neurons with the postsynaptic blockers AP5 and CNQX, we found that the neuronal response to ultrasound was not greatly affected (**Fig. 2j**), suggesting that synaptic transmission is not required for excitation. Synaptic transmission was shown to play a role in previous experiments ²⁵, and a small effect could not be ruled out by our results.*

Line 241: Differences between our findings and previous studies examining the response to FUS in vitro may be due to specifics of how cells were stimulated. For example, cells cultured on hard plastic or glass substrates may experience different mechanical and acoustic conditions than cells stimulated on top of acoustically transparent materials or within soft gels.

Other major issues need to be addressed are:

1. The comparison between 300 kHz and 670 kHz is interesting. The authors “found that

no significances in response amplitude or onset delay” in line 49 on page 4. However, in the caption of Supplementary Fig. 1g, the p value is described as 0.0011. Why is this comparison considered as not significant? What is the alpha level being used here?

→ Thank you for pointing this out. We misused the word “significant”, which should be reserved for statistical statements. We changed the wording to “substantial”. The alpha level used everywhere is 0.05.

2. A chirped waveform was used, and the authors mentioned in line 60 on page 4 that “we found the calcium signal unaffected in terms of response amplitude” as the data shown in Supplementary Fig. 4. The AUC may not be affected from the panel b, but it can be observed that the response onset delay is different. Is this difference statistically significant?

→ We thank the Reviewer for this great suggestion. We added new data in Fig. S4c showing the absence of a statistically significant increase in onset delay.

3. In lines 88-89 on page 6, the authors mentioned that “the acoustic radiation force on the neurons due to FUS is expected to be weak”. Any evidence to justify this claim?

→ This expectation arises from the similarity in acoustic impedance between the cells, the aqueous medium and the mylar substrate, and the absence of large acoustic field gradients within the ultrasound focus. We clarified this rationale in the text and added a reference to the relevant theory.

Line 92: Given the similar acoustic impedance of neurons and surrounding media and the absence of large pressure gradients within our ultrasound focus, the acoustic radiation force on the neurons due to FUS is expected to be weak^{43, 44}, with any resulting deformations expected to be below our optical detection limit.

4. In Fig. 4d and Supplementary Fig. 8b, the calcium responses, including the spontaneous activities, did not change significantly before and after blocking TRPV1, 2 and 4 channels using RR. As mentioned in the Methods, “minimum and working concentrations of chemical or peptide channel blockers were investigated by measuring the change of spontaneous calcium activities before and after applications”, how did the authors determine the minimum 1 μ M for RR? I wonder whether an increased RR concentration would lead to a significant change of calcium transients.

→ Thank you for this question. The IC₅₀ (half maximal inhibition concentration) of Ruthenium Red is around 200~500 nM⁵³, such that 1 μM RR should be sufficient to block TRPVs while avoiding non-specific inhibition of the neurons. We also tested a higher concentration (10 μM) and found non-specific inhibition. We added this new data in **Fig. S8b**.

5. Based on these in vitro experimental observations, it seems that the intramembrane cavitation theory and the NICE model are not valid/accurate to explain the ultrasound neuromodulation. Please further discuss this topic.

→ As suggested, we have expanded our discussion of this theory:

Line 248: Furthermore, no evidence was found to support a mechanism involving intramembrane cavitation and charge accumulation^{24, 29}, which predicts the formation of bubbles within a large fraction of the cell membrane, as well as strong hyperpolarization during FUS application. We observed no dependence of the neuronal response on solution degassing, which is expected to affect bubble dynamics, and our ultra-high-speed camera also did not record major changes in the appearance of the membrane, as might be expected with extensive formation of bubbles (whose refractive index differs from water). Furthermore, our imaging of membrane potential during FUS application did not reveal hyperpolarization. However, the limitations of our experimental tools prevent us from conclusively ruling out the intramembrane cavitation hypothesis: the optical resolution of our setup (> 500 nm) is insufficient to visualize very small bubbles, while our voltage indicator does not have sufficiently fast kinetics to observe voltage fluctuations on the ultrasound timescale.

6. Please discuss if the in vitro findings could be extended to in vivo systems. It would be nice if the authors could discuss this in light of recent in vivo reports.

→ As suggested, we have edited the Discussion to comment more on in vivo experiments:

Line 269: Indeed, recent studies are starting to examine the contributions of various cell types to the brain's response to FUS using intracranial electrical recordings^{28, 69-71}. These studies could be extended in the future by performing in vivo knockdowns of specific ion channels. Care must be taken to control for artifacts arising from the interaction of sound waves with recording electrodes and auditory side-effects^{30, 31}, which are not present in our in vitro preparation.

Minor issues and typos need to be clarified and fixed:

1. In Fig. 1b, what does the step waveform (above the raw ultrasound waveform) stand for?

→ We apologize the confusion. This step showed the timing of our command to the function generator (with the acoustic output delay representing sound propagation time). We don't think this step waveform is needed and have eliminated it from the figure.

2. In line 44 on page 4, the figure reference should be “Supplementary Fig. 1b”.

→ edited.

3. In the caption of Fig. 5h, please remove the redundant Δ .

→ edited.

4. In the lines 184-185 on page 12, the reference to Supplementary Fig. 7e seems incorrect here. Please check.

→ edited.

There was a typo: TTA-P2⁶¹ (IC50 \approx 400 22 nM, used at 3 μ M, Supplementary Fig. 7e).

5. In the line 203 on page 13, where is Fig. 7, h-j?

→ edited. Fig. 7, h-j f

6. In line 383 on page 19, change “...were treated...” to “...were used ...”.

→ edited.

7. The supplementary tables shared the same number and title. Please correct this.

→ edited.

8. The last column of Supplementary Table 2 should be “CFD score” instead of “CDF score”.

→ edited.

Reviewer #2 (Remarks to the Author):

In this manuscript, Yoo et al. dissect the mechanism of action by which focused ultrasound (FUS) excites neurons, through calcium influx and depolarization of the cell membrane. Their approach is based on calcium and voltage fluorescent imaging, combined to specific pharmacological and genetic inactivation of key candidate channels that might be involved in mediating the cellular response to FUS. The main parameters reported are the intensity and the delay of the calcium response upon FUS stimulation, and how they are impacted by channel modulations.

This elegant and effective approach manages to draw a convincing model for the excitation of cells after stimulation by FUS, where the mechanical stimulation of mechanosensitive calcium channels leads to the potentiation of sodium channels (such as TRPM4) that amplify the response and initiate a stronger, but delayed, calcium influx through voltage-gated calcium channels. This model is consistent with the response delay reported and provide major inputs on how to improve the kinetics of sonogenetics.

Overall, this manuscript is well written, the data is presented clearly, and the claims well justified by the careful experiments performed by the authors. Moreover, this kind of mechanistic work is incredibly valuable to the community who works on understanding new neuromodulation tools to create more effective techniques. This work will be an important foundational study for future efforts. It is a rare treat to encounter a manuscript as well-crafted as this one where it is difficult to find any major weaknesses. I want to congratulate the authors on this work, and I do not believe any additional experiments are required for publication. Nevertheless, I would be a poor reviewer if I did not offer suggestions for improvement. I believe these issues can be corrected by minor updates to the text. I do NOT think additional experiments are necessary at this point. These critiques are intended only to help strengthen and clarify this excellent work. Note I prepared these with significant help from a research scientist in the lab and we both agree on these critiques:

→ Thank you for this enthusiastic review and detailed suggestions, which helped improve the manuscript and are addressed below.

- We don't think a local increase in temperature at the membrane level can be entirely ruled out as a contributing factor based on the data presented here. Measurements done by optical thermometer measure bulk changes. Using RFP as a thermal reporter shows cytoplasmic changes with a sensitivity of 0.1 fluorescence per 5 °C, and only measures intracellular bulk temperature, not any local temperature on the membrane. It would be reasonable to assume that at the relevant timescales the temperature would be uniform throughout the cell, but this assumption should be specified in the text and justified if possible.

→ Thank you for this suggestion. We have updated the text to clarify the limitations of our temperature measurements and our interpretation of the results.

Line 72: Although these assays measure bulk temperature in media proximal to the cells or inside the cytoplasm rather than locally within the cell membrane, Fourier's law of heat diffusion predicts a thermal equilibration length scale on the order of 100 μm in aqueous media³⁹ during the 100 ms timescale of our observed responses. Our

results thus suggest that temperature is unlikely to play a major role in ultrasonic neuromodulation in this parameter range, as predicted by numerical estimates^{40, 41}.

- Similarly, membrane cavitation may not be entirely excluded as a contributing factor based on these data. Is it possible that the membrane can move to the extent that it may not be clearly visible from the microscopy data? What is the sensitivity of this experiment? Could you say that you can exclude cavitation larger than a specific displacement value? This should be made clear in the manuscript (results and discussion)

→ We agree with this comment and have updated our discussion accordingly.

Line 247: In addition, ultra-high frame rate imaging revealed no large-scale deformation or cavitation on the timescales of either the ultrasound cycle or the ultrasound pulse. Furthermore, no evidence was found to support a mechanism involving intramembrane cavitation and charge accumulation^{24, 29}, which predicts the formation of bubbles within a large fraction of the cell membrane, as well as strong hyperpolarization during FUS application. We observed no dependence of the neuronal response on solution degassing, which is expected to affect bubble dynamics, and our ultra-high-speed camera also did not record major changes in the appearance of the membrane, as might be expected with extensive formation of bubbles (whose refractive index differs from water). Furthermore, our imaging of membrane potential during FUS application did not reveal hyperpolarization. However, the limitations of our experimental tools prevent us from conclusively ruling out the intramembrane cavitation hypothesis: the optical resolution of our setup (> 500 nm) is insufficient to visualize very small bubbles, while our voltage indicator does not have sufficiently fast kinetics to observe voltage fluctuations on the ultrasound timescale.

- We would like to see a discussion on why there remains a 40% response in the presence of Gadolinium which should inhibit all mechanical response.

→ We appreciate this question. We believe this partial inhibition results in large part from our need to use a Gd³⁺ dose low enough to avoid off-target effects on neural excitability. At such a dose, the effects of Gd³⁺ on mechanoreception are only partial. In addition, some mechanosensitive channels do not require membrane deformation, and may therefore not be affected by Gd³⁺'s action at this dose. We have updated the text to make this clear:

Line 148: The incomplete elimination of the FUS response may be due to our use of a relatively low dose of Gd³⁺ to avoid non-specific effects, and possibly the involvement of mechanosensitive channels gated by mechanisms not requiring bilayer deformation⁵².

- Over-all, we think the response delay should be quantified throughout the text, for different conditions and different recording (calcium and voltage measurements) so that we can compare if the perturbations affect the delay.

→ We appreciate the Reviewer's suggestion, and have added several panels of new data to enable this additional comparison for critical experimental conditions. These additional figures are now mentioned at appropriate points throughout the text. Furthermore, we have uploaded

all the raw data from our experiments to allow interested readers to examine various quantitative aspects of the results.

Response delay for 2D vs 3D culture (Supplementary Fig. 3f).

Response delay for regular and chirped pulses (Supplementary Fig. 4c).

Response delay for cells treated with gadolinium, RR, GsMTx4 and suramin (Supplementary Fig. 8 f-i).

Response delay for CRISPR/Cas9 knockdowns (Supplementary Fig. 9, e-i).

- Discuss/explain: why do control recording have such different amplitudes (scales are different). In the CRISPR figure, specify what the control is (we are assuming it is a non-targeting sgRNA). Why are there variations in the response amplitude of non-targeting sgRNA between experiments? Why are the same controls used at times (ex: Fig. 5 d and e, or f and g). We are sure there are good reasons for why the controls are repeated in some experiments and not others, but this should be clarified for the reader.

→ Thank you for these questions. Our control is indeed a non-targeting sgRNA. We have clarified this in the caption of Fig. 5d and Fig. S9.

→ Variations in overall response amplitude arise due to the use of different neuronal preps on different experimental days, which was required to test this large number of experimental conditions. Neurons prepared on different days (obtained from a unique animal and transduced to express GCaMP) exhibited some variation in calcium responses. For each test condition, we therefore included a matching control from the same neuronal preparation. In some cases, we were able to run multiple conditions together on the same experimental day, in which case they shared a control. We added an explanation of this rationale in the Methods.

Line 363: Variations in overall response amplitude arise due to the use of different neuronal preps on different experimental days, which was required to test this large number of experimental conditions. Neurons prepared on different days (obtained from a unique animal and transduced to express GCaMP) exhibited some variation in calcium responses. For each test condition, we therefore included a matching control from the same neuronal preparation. In some cases, we were able to run multiple conditions together on the same experimental day, in which case they shared a control.

- Most traces: why show the sem rather than the individual traces to give a better sense of traces variability? Especially concerning delay. Perhaps a supplemental figure would be good here.

→ We agree that individual traces provide additional information, but also think the statistical plot of SEM provides a valuable quantification of variability. An example of both plotting approaches is provided below for comparison. We prefer the \pm SEM plots (which are commonly used in the literature) and would like to retain them, but welcome further guidance from the reviewer and editor. At the same time, we are sharing all our raw data so interested readers can examine single traces.

- We would be curious if FUS stimulation could be completely inhibited (other than by removing extracellular calcium), for example through multiple inhibition (drug cocktails). + Is there a dose dependence effect of major inhibitory drugs? Alternatively, since FUS doesn't work well on spiking HEKs, could response be improved by expressing missing channels that are essential in neurons' response to FUS?

→ Thank you for this comment. We agree that it would be interesting to test combinatorial inhibition in future studies. We did not conduct these experiments in the present study because of the implementation and interpretation challenges created by combinatorial effects of the drugs and their co-solvents (e.g., DMSO and methanol) on neuronal viability and spontaneous activity. We clarified this rationale in the text.

Line 162: Combinations of pharmacological blockers were not tested due to the potential combinatorial effects of the compounds and their co-solvents on neurons' viability and excitability.

We also agree that co-expressing a larger repertoire of channels in spiking HEK cells is an interesting topic for future studies, and have indicated this in the text.

Line 283: Furthermore, overexpression of the mechanoreceptors and amplifier channels identified in this study could sensitize cells that do not have intrinsic ultrasound responses, which would be of interest for both further mechanistic study (e.g., in spiking HEK cells) and the development of sonogenetic tools.

D. Appropriate use of statistics and treatment of uncertainties

T-test and Anova test are used appropriately when needed (except missing in figure 2b). However, it is not clear how the sem are calculated, are the authors using n as the number of dishes recorded, or the number or cells recordes (the grayed area indicating the SEM is often very small, hence my question). This should be clarified in the legends.

→ Thank you. We have clarified in the legends that n= is defined and SEM is calculated on the basis of dishes. We have also updated the methods section to clarify our averaging protocol.

Line 356: 50-300 cell bodies per each ROI (dish), depending on their seeding density, were detected by NeuroCa. Single-cell calcium signals from the ROI (dish) were averaged, and the averaged signals from each dish were used for data plotting. The number of biological replicates (n=) and \pm SEM were based on the number of dishes.

We also added the missing statistics in Fig. 2b: (Unpaired T-test, $p < 0.0001$). Sorry for this omission.

F. Suggested improvements: experiments, data for possible revision

- For traces and area under the curve, for each FOV (for each dish) it is not clear if the fluorescence of each cell is measured, then averaged, or if the fluorescence of the entire FOV is extracted. Please clarify in methods or in legends.

→ Thank you. We clarified this issue in response to the preceding comment.

Reviewer #3 (Remarks to the Author):

This manuscript under review describes an in depth study into the biophysical and molecular mechanism of ultrasound neuromodulation in an in vitro primary neuron system, and proposes a plausible hypothesis of mechanosensitive channel activation and calcium amplification. Ultrasound neuromodulation has been an emerging field with great clinical potential, yet the detailed mechanism remains elusive. This manuscript provides important new insight to the existing literature. The study itself is very thorough and clearly laid out, including dose and frequency dependency of ultrasound modulation, pharmacological inhibition of multiple mechanosensitive ion channels and the results are supported by genetic modifications. In addition, other hypothesis such as temperature and intramembrane cavitation has been carefully ruled out. Given the strong promise of this work, I suggest publication after revisions.

→ Thank you for your positive review and helpful suggestions, which are addressed in detail below.

The average power of acoustic wave used in this study was 15W/cm², which seems to be higher than that used in other studies (e.g. 0.228W/cm² in <https://doi.org/10.1016/j.neuron.2010.05.008>).

The activation latency of 200 ms described in this manuscript seems to contradict previous reports of US induced EMG response with delay of less than 100ms. (<https://doi.org/10.1038/s41467-021-22743-7>, <https://doi.org/10.1016/j.brs.2019.03.005>)

What are the possible factors contributing to this delay?

→ We appreciate this comment. We focused on acoustic intensities within the range of most large animal and human studies, seeing substantial excitation down to 3W/cm² (Fig. 1d). We also ensured that the acoustic parameters used are not damaging to the cells. This is clarified in the text on line 51:

*Line 51: Based on these results, we set our subsequent stimulation parameters to 15 W/cm² and 500 ms (CW), which are similar to those used in large animal and human studies^{9, 11, 14-19, 32}. To ensure that these ultrasound parameters were not damaging to cells, we looked for and found neither sustained calcium accumulation nor irreversible membrane perforation after repetitive stimuli (**Supplementary Fig. 2**).*

While some *in vivo* studies have observed responses to lower intensities and with shorter latencies, some of these studies positioned electrodes in the ultrasound focus (altering the mechanics of the system) and/or did not account for indirect auditory effects, which may occur on a faster timescale.

In fig3d, the voltage trace of the neurons do not resemble firing of action potentials as shown in fig S6b. Is this a subthreshold response? Are there trials where action potentials were observed after initial depolarization?

→ Thank you for this question. Time scale of the voltage trace in Fig. S6b is longer than Fig. 3d, which may cause confusion. We updated the voltage trace in Fig. S6b to re-scale it as in Fig. 3d. Furthermore, the voltage trace in Fig. S6b was recorded from a dish with 100 Hz recording speed, while the trace in Fig. 3d was an averaged signal from 4 dishes in which we

used a higher recording speed (200 Hz) to help measure response latency. With this frame rate (and correspondingly reduced SNR), we were not able to capture clean traces from individual cells. This is why the two types of voltage recordings look different, with the data in Fig. 3d capturing average firing responses rather than distinct spikes (resulting in a smaller maximum observed % change in fluorescence). We have clarified this point in the Methods.

Line 375: For voltage imaging of neurons, we used a higher recording speed (200 Hz) to observe the response latency. With this speed (and correspondingly reduced SNR), we were able to capture average firing responses from multiple dishes rather than distinct spikes from individual cells.

Replotted spontaneous activity trace (Supplementary Fig. 6b).

In fig 4a, in addition to block mechanosensitive ion channels, Gd will also alter all membrane mechanical properties by increasing membrane rigidity so that other possible mechanisms, if any, will also be suppressed.

→ Thank you for this comment. We agree that Gd^{3+} can cause a broader range of effects on membrane mechanics, and have clarified in the revised main text that our use of Gd^{3+} is intended to examine the role of mechanical deformation in the neuronal response to ultrasound rather than the role of specific mechanosensitive channels. We also clarified that we are using Gd^{3+} at a concentration that does not more globally inhibit neuronal excitability (which can happen at higher concentrations).

*Line 143: We started by treating the neurons with gadolinium(III), which modifies the deformability of the lipid bilayer⁵², resulting in changes in membrane mechanics leading to inhibition of mechanosensitive ion channels. The dose of Gd^{3+} was carefully chosen to avoid blocking non-mechanosensitive channels or otherwise altering cell excitability⁵³ (**Supplementary Fig. 8a**).*

In Fig 5cde, why do the calcium traces go below the baseline after stimulation?

→ Thank you for this question. Baseline fluctuation of GCaMP6f has been observed in other *in vitro* and *in vivo* studies (Badura et al., Neurophotonics, 1(2), 025008 (2014), Chen et al., Nature, 499, 295–300 (2013)), and it can also be seen with other versions of GCaMP (Dana et al., Nature Methods 16, 649–657 (2019)). This is why we are not surprised to see a few traces going below baseline in some of our experiments.

→ Thank you for your helpful review!

Reviewers' Comments:

Reviewer #1:

Remarks to the Author:

The authors have addressed all my comments and the revision has been much improved. This is an outstanding and timely contribution to the field.

Reviewer #2:

Remarks to the Author:

The authors have addressed the concerns we had and I recommend this manuscript for publication. I do however recommend the authors plot the calcium dynamics with individual traces rather than +/- sem. The individual responses are much more reproducible than I had expected and think this is valuable information for the reader that does not compromise the ability to read the data.

Reviewer #3:

Remarks to the Author:

The authors have addressed my comments. I recommend publication.

Response to Reviewer Comments

Reviewer #1 (Remarks to the Author):

The authors have addressed all my comments and the revision has been much improved. This is an outstanding and timely contribution to the field.

Reviewer #2 (Remarks to the Author):

The authors have addressed the concerns we had and I recommend this manuscript for publication. I do however recommend the authors plot the calcium dynamics with individual traces rather than \pm sem. The individual responses are much more reproducible than I had expected and think this is valuable information for the reader that does not compromise the ability to read the data.

→ We re-plotted the calcium dynamics with individual traces (in addition to the \pm SEM) in Fig 1d and e to provide an example of the relative uniformity of the responses.

Calcium responses as function of ultrasound intensity and pulse duration (Fig. 1, d,e)

Reviewer #3 (Remarks to the Author):

The authors have addressed my comments. I recommend publication.